# Deconstruction by *C. thermocellum*—from microbe mediated to dynamic redistribution of cellulosomes

John M Yarbrough[1], Neal N Hengge[1], Qi Xu[1], Samantha J Ziegler[1], Daehwan Chung[1], Shu Huang[1], Sarah Moraïs[2], Itzhak Mizrahi[2], Edward A Bayer[2,3], Yannick J Bomble[1]

*Clostridium thermocellum* is one of the most efficient microorganisms for the deconstruction of cellulosic biomass. To achieve this high level of cellulolytic activity, *C. thermocellum* uses large multienzyme complexes known as cellulosomes to break down complex polysaccharides, notably cellulose, found in plant cell walls. The attachment of bacterial cells to the nearby substrate via the cellulosome has been hypothesized to be the reason for this high efficiency. The region lying between the cell and the substrate has shown great variation and dynamics that are affected by the growth stage of cells and the substrate used for growth. Here, we used both super-resolution imaging and machine-learning approaches to study the distribution of *C. thermocellum* cellulosomes at different stages of growth. We show that *C. thermocellum* initially retains its cellulosomes primarily on the cell surface but then relocates large cellulosome clusters to the interface with biomass, therefore depleting its cell surface of cellulosomes. These results indicate dynamic redistribution of cellulosomes during growth, with a functional shift toward substrate-associated degradation later during growth on biomass.

## Introduction

Many microorganisms have evolved different mechanisms to efficiently deconstruct lignocellulosic biomass (1). To date, *Clostridium thermocellum*, a thermophilic anaerobe, is the most efficient known biomass degrader found in the biosphere with the capacity to solubilize more than 55% of the polysaccharide content of many key biomass feedstocks (2). It has been hypothesized that this high propensity for biomass deconstruction is due to its complex and sophisticated cellulolytic machinery, which relies on supramolecular multienzyme complexes called cellulosomes (3, 4, 5, 6, 7). These cellulosomes comprise nonenzymatic scaffolding proteins (scaffoldins) decorated with a wide variety of complementary polysaccharide-degrading enzymes. In this cellulosomal system, complex formation is mediated by a strong interaction between the type I dockerin module of the enzymes and the complementary type I cohesin module found on the scaffoldins. The primary scaffoldin in *C. thermocellum*, CipA, is composed of nine type I cohesin modules, which can bind nine type I dockerin-bearing enzymes, a type II dockerin module, and a family 3 carbohydrate-binding module (CBM3a) (Fig 1) (8, 9). These scaffoldins can themselves be assembled onto secondary scaffoldins bearing type II cohesin domains, specifically through interactions with three critical anchor proteins: the heptavalent OlpB, Orf2P, and SdbA. Each of these anchor proteins contains a cohesin domain that binds with high specificity to the type II dockerin present on the primary CipA scaffoldin. OlpB, Orf2P, and SdbA not only facilitate the hierarchical assembly of a supramolecular complex that can incorporate up to 63 glycoside hydrolase enzymes but also anchor this complex firmly to the microbial cell surface via their surface layer homology (SLH) domains. The SLH domains provide attachment points within the bacterial cell wall enhancing proximity to the microbial cell. This structural organization and surface localization confer substantial advantages to cellulosome-producing microbes such as *C. thermocellum*, enabling them to degrade lignocellulosic biomass more efficiently compared with organisms that rely on free cellulases (9, 10, 11, 12, 13). Moreover, the CBM3a allows the scaffoldin to selectively bind to cellulose. As a result of these varied interactions, *C. thermocellum* has been shown to adhere very strongly to the biomass substrate (14, 15).

*C. thermocellum* has been extensively studied over the years to address many scientific questions regarding this elaborate cellulolytic system. Critical questions raised include the following: (1) What is the importance of the scaffoldins and the attachment of the cellulosome to the microbe? (2) How do cellulosomes and their composition change during growth, and are they released into the media?

Given the extensive literature regarding *C. thermocellum* and cellulosomes, we can now paint a clearer, albeit still incomplete,

[1]National Renewable Energy Laboratory, Bioscience Center, Golden, CO, USA   [2]Department of Life Sciences and the National Institute for Biotechnology in the Negev, Ben-Gurion University of the Negev, Beersheba, Israel   [3]Department of Biomolecular Sciences, The Weizmann Institute of Science, Rehovot, Israel

Correspondence: yannick.bomble@nrel.gov

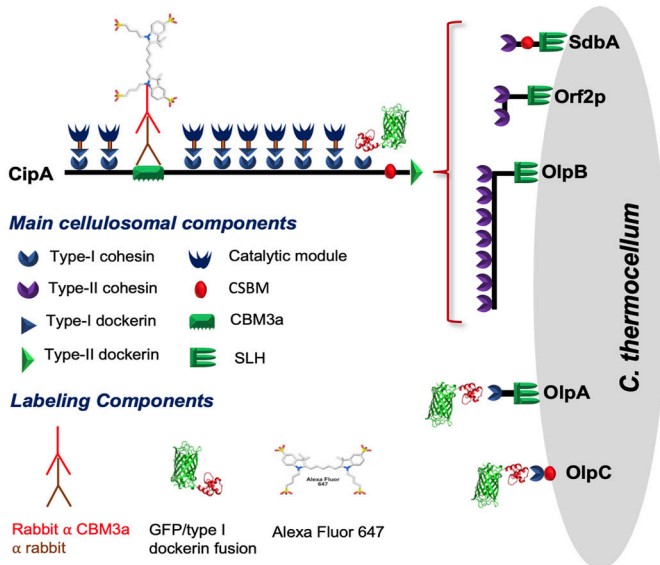

**Figure 1.   Main cellulosomal system of *C. thermocellum*,** including cell-bound cellulosomes and components of our labeling strategy, using a combination of antibody-targeted chromophores specific to the family 3 carbohydrate-binding module (CBM3a) and a green fluorescent protein-dockerin fusion protein (GFP-dockerin) along with the surface layer homology (SLH) domain and the cell surface binding module (CSBM).

the microbe and biomass surfaces. This reactive region has been hypothesized to be a key to the high cellulolytic activity of this microbe.

Several approaches can be used to study the EMS interface between *C. thermocellum* and biomass, such as scanning electron microscopy (SEM) or transmission electron microscopy (TEM) with immunolabeling. The latter, for example, was used by our group to characterize these interactions with limited success. Both conventional TEM and SEM can technically achieve the resolution needed to study the cellulosome, with resolutions around 0.2 nm (18, 19) and 2 nm (20, 21), respectively. However, the primary drawback of these techniques for this application is the extensive sample preparation needed, which can lead to long time lags between the main experiments and imaging of the sample. Furthermore, there is a lack of reliable statistics to analyze the level of immunolabeling because of quantification difficulties. Optical microscopy can help alleviate the need for cumbersome sample preparation and improve quantitative analysis. However, until recently, the resolution of these techniques (greater than 250 nm in most cases) was restricted because of the diffraction barrier of light (22, 23). Nevertheless, within the last 10 yr, a new field of optical microscopy has emerged that is capable of capturing the location of single fluorescent molecules with an optical resolution between 10 and 30 nm rendering them particularly well suited for more accurate localization of target molecules under near-native conditions (22, 24, 25) (see Supplemental Data 1).

Here, we used both photoactivation localization microscopy (PALM) (26) and stochastic optical reconstruction microscopy (STORM) (24, 27) to study the distribution of *C. thermocellum* cellulosomes at different stages of growth when actively growing on soluble and insoluble substrates, which contributes toward a clearer picture of the dynamics of cellulosome populations at the EMS interface. We show in this work that (1) *C. thermocellum* bacteria grown on soluble substrates exhibit large cellulosome clusters in the log phase that become smaller over the course of growth, (2) the concentration of cellulosomes in the EMS region, when bacteria are actively engaged on insoluble substrates, is far greater than previously expected, (3) bacteria actively bound to insoluble substrates increase the local concentration of cellulosomes at the contact points, and (4) bacteria that are not bound to the substrate (most likely after leaving the substrate) are depleted in surface-attached cellulosomes.

picture of the elaborate functioning of its cellulolytic machinery. For example, it has been shown that the bacterial cell walls are decorated with cellulosomes by Bayer and Lamed who were the first to show protuberance-like nodules on the outer surface of the bacterial cell walls using transmission electron microscopy (TEM) (14). In the same work, they also showed that the conformation of cellulosomes changed when engaged on cellulose. Furthermore, Bomble and coworkers were able to demonstrate, using molecular modeling, that even though dockerin–cohesin interactions seem to be similar for all cellulosomal glycoside hydrolases, the assembly process on the CipA scaffoldin was not random, but rather depended on the size and flexibility of these biomass-degrading enzymes (16). This notion garnered experimental support by evaluating the incorporation of different cellulosomal enzymes into a trivalent mini-scaffoldin (17). More recently, Xu and coworkers took steps to develop several genetically modified strains of *C. thermocellum* to selectively delete the different scaffoldins that make up its cellulosome system, including those that tether cellulosomes to the bacterial cell wall (9). Overall, these genetically modified strains were still able to degrade crystalline cellulose and biomass with the exception of the CipA-deleted strain, which led to an almost complete inactivation of cellulolytic activity of the microbe (9). Thus, the presence of the primary CipA scaffoldin is intrinsic to the observed efficient degradation of cellulosic biomass. Interestingly, some of these deletions also led to a change in the biomass deconstruction mechanism used by this microbe. The deconstruction mechanism of WT *C. thermocellum*, which proceeds via defibrillation of the biomass, can be attributed to its attachment to the biomass surface and, notably, the dynamics of the so-called enzyme microbe substrate (EMS) interface located between

## Results

### Establishing the use of super-resolution imaging of *C. thermocellum* on different substrates using multiple probes

Early studies have reported that the presence of free cellulosomes on the surface of cellulose may be related to the detachment of the bacterium from the substrate (8, 9, 14, 15). These studies also noted a reduction of cell-associated cellulosomes in the stationary phase of cell growth compared with the log phase. It was further proposed that detachment of the bacterial cells may be connected to a controlled release of the cells from the cellulose-bound

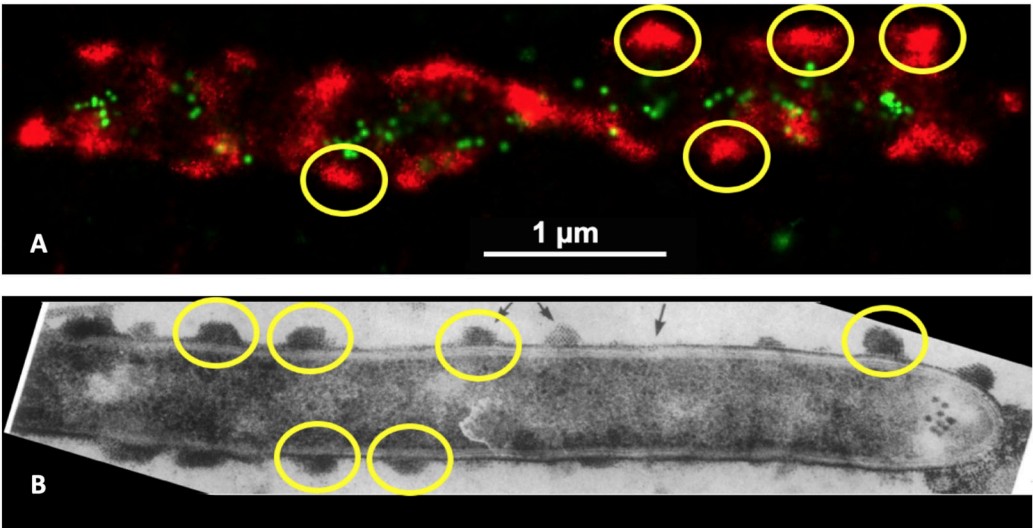

**Figure 2. Distribution of cellulosomes on the surface of *C. thermocellum*.**
**(A, B)** Super-resolution optical image of a *C. thermocellum* cell with the CBM3a located on the CipA scaffoldin tagged with Alexa Fluor 647 (red fluorescence) and a PA-GFP type I dockerin fusion protein (green fluorescence) associated with type I cohesins of either CipA, OlpA, or OlpC scaffoldins and (B) adapted figure from Bayer et al *Ultrastructure of the cell surface cellulosome of Clostridium thermocellum and its interaction with cellulose* showing a transmission electron micrograph of *C. thermocellum*, labeled with cationized ferritin (15). In these images, some of the cellulosome protuberances are highlighted with yellow circles, demonstrating that the surface of the bacterium is decorated with the populated CipA scaffoldin.

cellulosome, allowing the continuation of hydrolysis of the substrate without the need for the bacterial cell to be present on the substrate. Moreover, a recent study (reference 28) examined single *C. thermocellum* cells at nanoscale resolution at different growth phases using cryo-ET, and observed phenotypic cellulosomal heterogeneity regulated by soluble sugar concentration in the media, thus suggesting a division-of-labor strategy. We therefore sought to clarify further the status of the cellulosome with regard to the presence of both the primary scaffoldin and the cellulosomal enzymes on cells of *C. thermocellum* and on the substrate during the different phases of growth, using advanced optical microscopic approaches.

To enable the imaging of different components of the cellulolytic machinery of *C. thermocellum*, we used a primary antibody targeting the CBM3a of the CipA scaffoldin and a secondary antibody fused to Alexa Fluor 647 (27, 29), in combination with an engineered photoactivatable GFP (PA-GFP) (30), fused to a type I dockerin (Fig S1) that specifically targets type I cohesins found on the CipA, OlpA, and OlpC scaffoldins, for STORM and PALM imaging, respectively (Fig 1). Fig 2A shows the fluorescence of these two probes in conjunction with *C. thermocellum* cells grown on cellobiose in the stationary phase of growth. These super-resolution optical images are reminiscent of the work conducted by Bayer and coworkers (Fig 2B) wherein transmission electron micrographs of *C. thermocellum* showed similar protuberance-like structures (15). Bayer and coworkers discovered these protuberances on *C. thermocellum* and reported that both their size and longitudinal arrangement varied as demonstrated by either a selective cationic electron-dense probe or antibodies specific to the CipA scaffoldin (15). Using STORM and PALM imaging, we were able to further validate these original findings using minimal sample preparation and disruption of the bacterium with a resolution of 35 nm (Figs S2

and S3). To date, these cellulosomal protuberances and their dynamics have not been fully characterized, especially on process-relevant lignocellulosic biomass. These advanced optical microscopy techniques appear to be particularly well suited to the quantitative study of the population of cellulosomes and their dynamics over time during biomass deconstruction.

In this context, Fig 3 shows the distribution of cellulosomes (as indicated by the presence of CBM3a) on *C. thermocellum* in the log phase (Fig 3A and B) and stationary phases (Fig 3C and D) when grown on cellobiose (additional images of these bacterial cells can be found in Fig S4). From these data, there is a distinct change in the pattern of the polycellulosomal protuberances, which form highly decorated and interconnected protuberances on the bacterial cells in the log phase. These new structures are absent when the microbe is in the stationary phase of growth. The fused GFP-dockerin probe also allows us to visualize the empty type I cohesin positions on the primary scaffoldins CipA, as well as those on scaffoldins OlpA and OlpC that are tethered to the cell surface (Fig 1). Surprisingly, most of the empty type I cohesins are located on the bacterial cell wall.

To determine whether the behavior of cellulosomes was different when *C. thermocellum* was grown on insoluble substrates, we grew *C. thermocellum* on microcrystalline cellulose (Avicel) particles and imaged the bacteria and substrate, 24 h after inoculation, using the aforementioned probes. Fig 4 shows the fluorescence corresponding to the CipA-CBM3a antibody and the GFP/type I dockerin fusion construct with the bacterial cells clearly attached to the Avicel particles. Most cells appear to have fewer cellulosome clusters than in the log-phase cells, grown on cellobiose, except at the EMS interface. In addition to the red and green fluorescence of the CBM3a and the scaffoldins, respectively, the yellow fluorescence is indicative of a colocalization of the

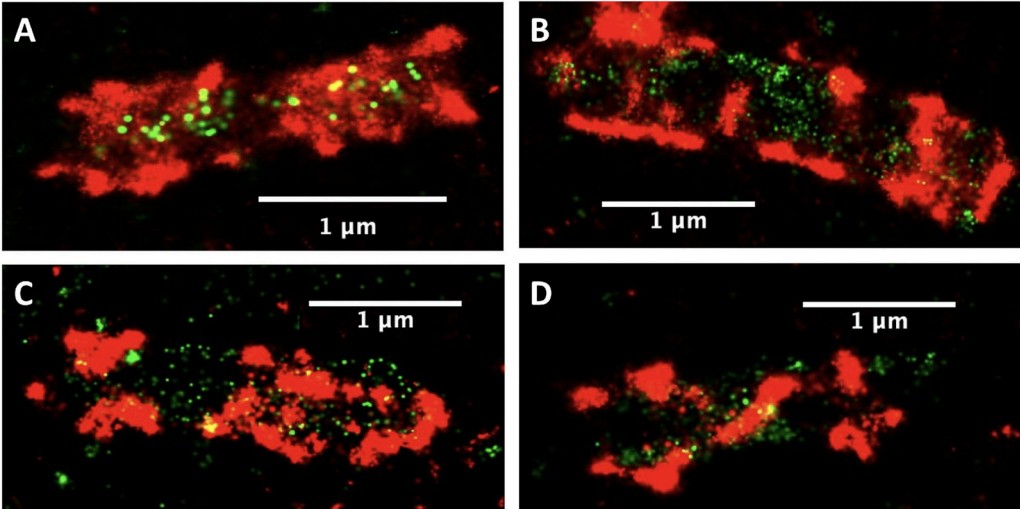

**Figure 3.** *C. thermocellum* **grown in the presence of cellobiose in the exponential growth phase.**
**(A, B, C, D)** *C. thermocellum* grown in the presence of cellobiose in the exponential growth phase (A, B) with an increase in the number of cellulosomes decorating the bacterial cell wall, in contrast to bacterial cells captured in the stationary phase (C, D) wherein the CipA scaffoldins are still present but with significantly reduced abundance. Red fluorescence denotes CBM3a (CipA scaffoldin)-associated label, whereas green fluorescence indicates unoccupied type I cohesin–related sites (see the legend to Fig 2 and text for more detail). The panels in this figure represent a subset of those shown in Fig S4.

green and red fluorescent molecules, signifying that there is a CBM3a in close proximity to an unoccupied type I cohesin.

The distribution of fluorescence on the Avicel particle indicates that there is a significant increase in CBM3a and unoccupied type I cohesins at the EMS interface when the bacterium is attached to Avicel. The same observation can be made to some degree for cells binding to Avicel particles and on the rest of the particle itself (in the absence of bacterial cells). A similar increase in the fluorescence within this region was observed on other Avicel particles (see Supplemental Data 1), thus signifying that this high concentration of cellulosomes is not unique. Along with the increase in concentration of cellulosomes within the EMS interface, there is also a substantial amount of colocalization (yellow fluorescence) compared with cells grown on cellobiose, pointing to unoccupied (i.e., lacking type I dockerin-bound enzyme), type I cohesins (most likely from CipA) adjacent to CipA-CBM3a. Also, the fluorescence pattern on the bacteria seems to show two states, one in which the bacteria are decorated with CipA clusters (red arrows in Fig 4A) and one in which the bacteria have few CipA clusters (white arrows in Fig 4A) with lower fluorescence intensity, most likely indicating either that they are in different stages of growth or that they may have already relocated their cellulosomes to the contact points. Fig 4B shows just the fluorescence form the Alexa Fluor 647 (CBM3a) and the photoactivated GFP/type I dockerin fusion protein to help highlight the difference in intensity between the more decorated bacteria versus the bacteria with low fluorescence clusters (additional images of these bacterial cells can be found in Fig S5). These results agree with previous observations of phenotypic heterogeneity within *C. thermocellum* populations that were connected to a division-of-labor strategy (28).

In this section, we have shown that we can efficiently image different components of the *C. thermocellum* cellulolytic machinery using STORM and PALM on soluble and insoluble

substrates with a resolution of close to 30 nm (Fig S2). These results clearly exemplify how powerful these techniques can be to answer many outstanding scientific questions about biomass deconstruction by *C. thermocellum*. However, current approaches to analyze super-resolution images do not take advantage of the abundance of data that these techniques can provide. To remedy this problem, more quantitative approaches are needed to enable the high-throughput analysis of tens to hundreds of bacterial cells.

## A promising unsupervised machine-learning approach

Traditional far-field optical microscopy, even with automated analysis tools, is limited by the diffraction limit of light, which prevents precise quantification of individual fluorophores within bacterial cells. This limitation causes averaging of fluorescence signals, loss of molecular detail, difficulty accounting for heterogeneity, and susceptibility to background, autofluorescence, and photobleaching effects, therefore restricting our ability to quantify the fluorescence from each bacterium, let alone analyze any significant portion of the bacterial cells. Super-resolution methods like STORM and PALM overcome these challenges by localizing and counting individual fluorophores with nanometer precision, enabling accurate quantification, high-resolution mapping of molecular distributions, and reduced out-of-focus light interference. Because a portion of the data generated from the super-resolution images contains the x-y coordinates of the calculated Gaussian for each fluorescent event, we can use this information to quantify the location and number of fluorescent events within a region of the image using unsupervised machine learning. This type of machine learning, also referred to as cluster analysis, allows us to define a certain epsilon radius (Eps) and set a minimum number of points, or, in this case, the number of fluorescent molecules within the defined radius. The clustering algorithm chosen for this work was

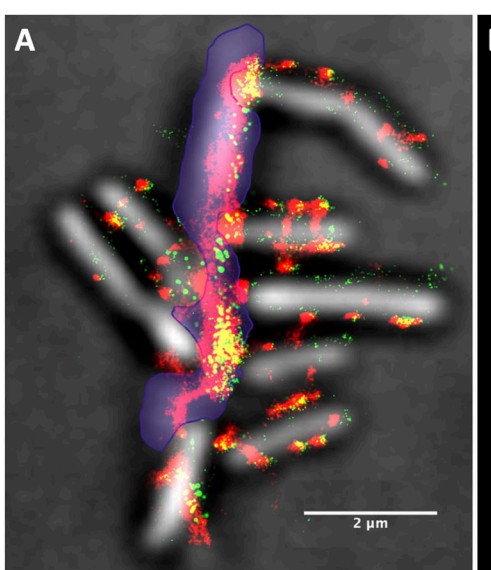
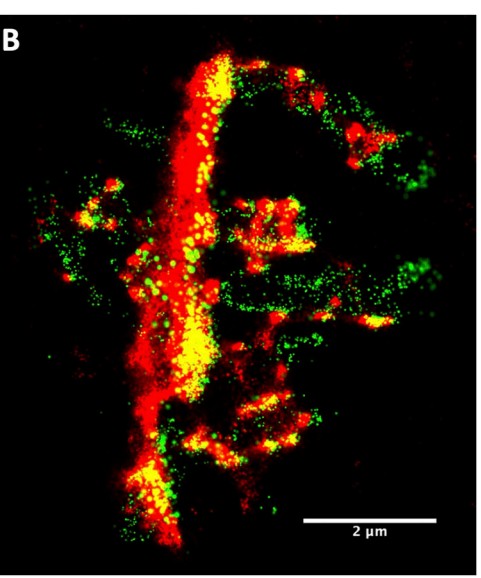

**Figure 4. *C. thermocellum* grown on Avicel in stationary phase.**
**(A, B)** Images of *C. thermocellum* grown on Avicel captured in the stationary phase showing (A) the white-light image superimposed with the fluorescence from the Alexa Fluor 647 (CBM3a) and the photoactivatable GFP/type I dockerin fusion protein and (B) only the fluorescence from the Alexa Fluor 647 (CBM3a) and the fluorescence from the photoactivatable GFP/type I dockerin fusion. **(A)** Avicel particle is highlighted with purple shading in (A). The increase in fluorescence within the EMS interface (area where the bacteria are connected to the Avicel) signifies that there is a high concentration of cellulosomes in that region.

the density-based spatial clustering of applications with noise (DBSCAN), which is a propagative cluster detection method linking points that are closely packed together and detecting outliers using the user-defined epsilon radius and a minimum number of points or molecules (31). Some of the main benefits of using DBSCAN for single-molecule experiments are that DBSCAN can detect arbitrary-shaped clusters, is quick, and is robust to outliers (31). For the work presented here, we initially decided to set the epsilon radius to 75 nm for the CBM3a-tagged Alexa Fluor 647 molecules.

Fig 5 illustrates the DBSCAN algorithm with the green spheres representing the core points considered part of a cluster as they fulfill two criteria: (1) they are located within the epsilon radius and (2) they contain at least the minimum number of molecules (MNM). The epsilon radius is defined as the maximum center-to-center distance within which two localized molecules are considered spatial neighbors. In our analysis, we set the epsilon radius to 75 nm, a value chosen to reflect the estimated upper bound of distances between one CBM3a-binding sites within a single cellulosome and another CBM3A-binding sites within another single cellulose based on structural models and to account for localization precision, allowing for a direct physical interpretation in the context of cellulosome organization. For the MNM parameter, this parameter specifies the smallest number of neighboring molecules within the epsilon radius to classify the localization as core molecules. The value of the MNM was selected empirically, taking into account the labeling density and localization uncertainties. Molecules that meet both the epsilon radius and the MNM criteria are designated as core molecules of the same cluster.

During the DBSCAN algorithm, if a cluster is started based on the described criteria of the epsilon radius and the MNM, the algorithm will continue classifying each point and propagate the cluster, while each scanned point meets the given criteria for epsilon radius and the MNM. Once one of the criteria is not met, these

points are classified as the border points (border molecules, orange spheres in Fig 5). If a point does not meet the minimum number of points within the defined epsilon radius, then the point is classified as noise (purple spheres in Fig 3).

## *C. thermocellum* sheds cellulosomes over the course of growth on cellobiose

Fig 6 shows the distribution of cellulosomes (as indicated by the presence of CBM3a) on *C. thermocellum* in the log and stationary phases when grown on cellobiose. From these data, there is a distinct change in the pattern of the polycellulosomal protuberances, which form highly decorated and interconnected protuberances on the bacterial cells in the log phase. These new structures are absent when the microbe is in the stationary phase of growth. The fused GFP-dockerin probe also allows us to visualize the empty type I cohesin positions on the primary scaffoldins CipA, as well as those on scaffoldins OlpA and OlpC that are tethered to the cell surface (Fig 1). Surprisingly, it seems that most empty type I cohesins are located on the bacterial cell wall and that the CipA scaffoldins are almost fully saturated with enzymes. To get a better understanding of the distribution of the CipA scaffolds, DBSCAN cluster analysis was used to analyze the images using a combination of the number of clusters per bacterial cell versus the MNM per epsilon radius.

A visual example is given with the number of clusters and their overall size on a bacterial cell in the log phase and in the stationary phases of growth. In this example, the number of cellulosome clusters found on the cell in the stationary phase with an MNM per epsilon radius of 200 was four clusters (6D), and the number of clusters found for the bacterial cell in the log phase for 200 MNM per epsilon radius was nine (6C). When we increased the MNM per epsilon radius, in both cases, we saw a decrease in the number of detected nodules. For example, for 500 MNM per epsilon radius (Fig 6E and F), two nodules were detected in the stationary phase,

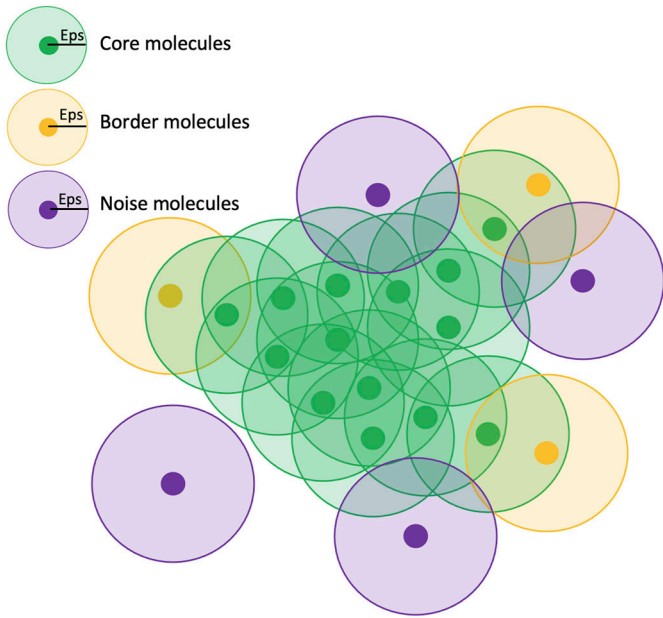

**Figure 5. Graphical representation showing how the DBSCAN algorithm defines a cluster.**
The green spheres represent the core groups within the cluster, because they contain the minimum number of points within the defined Eps radius. The orange spheres represent the border points, which are still part of the cluster because they are within the defined epsilon radius but do not meet the minimum number of points criteria. The purple spheres indicate noise points and are not assigned to a cluster.

but eight were detected in the log phase, thus illustrating that the nodules on the bacterial cells are larger in the log phase.

Altogether, we analyzed the normalized numbers of clusters collected from over 60 bacterial cells in both the log and stationary phases of growth with an example of one bacterial cell in the log phase (Fig 6A) and the stationary phase (Fig 6B). Note that the x-axis in Fig 6G and H represents the MNM per epsilon radius but does not give the relative size of the cluster, and the y-axis in Fig 6G and H represents the normalized number of clusters per bacterial cell. Normalizing the number of clusters per bacterial cell gains quantitative insight into how the density of molecules within the cellulosomal protuberances is affected during growth. During the log phase (purple bar in Fig 6G and dark green in Fig 6H), there is an increase in the density of molecules compared with that of bacterial cells in the stationary phase. In the stationary phase (yellow bar in Fig 6G and light green bar in Fig 6H), the number of identified clusters decreases as the density of molecules increases starting from 100 molecules per epsilon radius, whereas the maximum number of clusters per bacterial cell in the log phase is associated with 300 molecules per epsilon radius. This analysis was also used to determine potential clustering of unoccupied type I cohesins on microbial cells (Fig 6H). The fluorescence corresponding to unoccupied cohesins shows limited clustering overall with only one cluster with 10 molecules per epsilon radius. Most importantly, the clustering of unoccupied cohesins on the *C. thermocellum* bacterial wall does not seem to follow the same pattern as that of the CipA-CBM3a, as the distribution in both the log phase (from

67 single bacterial cells (Fig S4)) and the stationary phase (from 70 single bacterial cells (Fig S5)) remains the same in both stages of growth. The signal for the unoccupied type I cohesins and these analyses indicate that they are fairly dispersed indicating either a low abundance of unoccupied type I cohesins or a low abundance of type I cohesins on the surface of these bacteria.

### *C. thermocellum* relocates cellulosomes over the course of growth when attached to insoluble substrate

To study the distribution of cellulosomes on bacterial cells and substrate during biomass deconstruction, we subjected *C. thermocellum* cells grown on Avicel (representative of an insoluble substrate) to the same imaging and analysis procedure. The bacterial cells were again captured during two stages of growth, log and stationary. Because of the added complexity of an insoluble substrate, each image set was inspected, and a python algorithm was used to select either individual bacterial cells, multiple bacterial cells (if the cells were too close in proximity to be captured individually), Avicel particles, or Avicel particles and bacteria together (see the Materials and Methods section for more details on this procedure).

Fig 7 shows a collection of images corresponding to *C. thermocellum* cells actively degrading Avicel during logarithmic growth starting with the original image (Fig 7A) and the specific region of interest in Fig 7B. The Avicel particle in this figure is covered with multiple bacteria and identified as the underlying speckled red fluorescence with less intensity than the bacteria themselves (Fig 7C). This analysis shows the extensive network of CipA scaffoldins (cellulosomes) on both the bacterial cell wall and the Avicel particle (Fig 7D), whereas Fig 7E and F shows all unoccupied type I cohesins on either the CipA scaffoldin or the bacteria (OlpA or OlpC scaffoldins). From this image, the complexity of the CipA scaffoldin network on both the surface of the bacteria and on the surface of the Avicel particle is quite visible and makes it rather difficult to draw any conclusion by simply comparing one section of the image with another. Therefore, these data, along with 15 other Avicel particles and 75 individual bacterial cells, were analyzed using the DBSCAN clustering algorithm. The clustering algorithm ran through an array of MNM per epsilon radius from 10 to 1,200 at a fixed Epsilon radius set at 75 nm.

The results of this analysis with the MNM set to 40 for the CBM3a (Fig 7D) show a total of 205 clusters of CipA-CBM3a identified, in which most of the clusters are located on the bacterial cell wall, thus demonstrating that there are high concentrations of CipA scaffoldins on the surface of the bacteria. It is very surprising to have more than 40 CipA scaffoldins packed in an area of ~0.018 square microns, meaning that cellulosomes can be very tightly packed on the surface of the bacterium. For an MNM set to 500, there are a total of 127 clusters, again suggesting that cellulosomes can be very tightly packed within a small area, and these clusters are still mainly found on the surface of the bacterial cell wall. The cluster analysis for the PA-GFP tagged to the type I dockerin with the MNM set to 20 with an epsilon radius of 125 nm led to the identification of 263 clusters. PA-GFPs are associated with either an unoccupied OlpA/OlpC site located on the bacterial cell wall or an unoccupied type I cohesin located on the

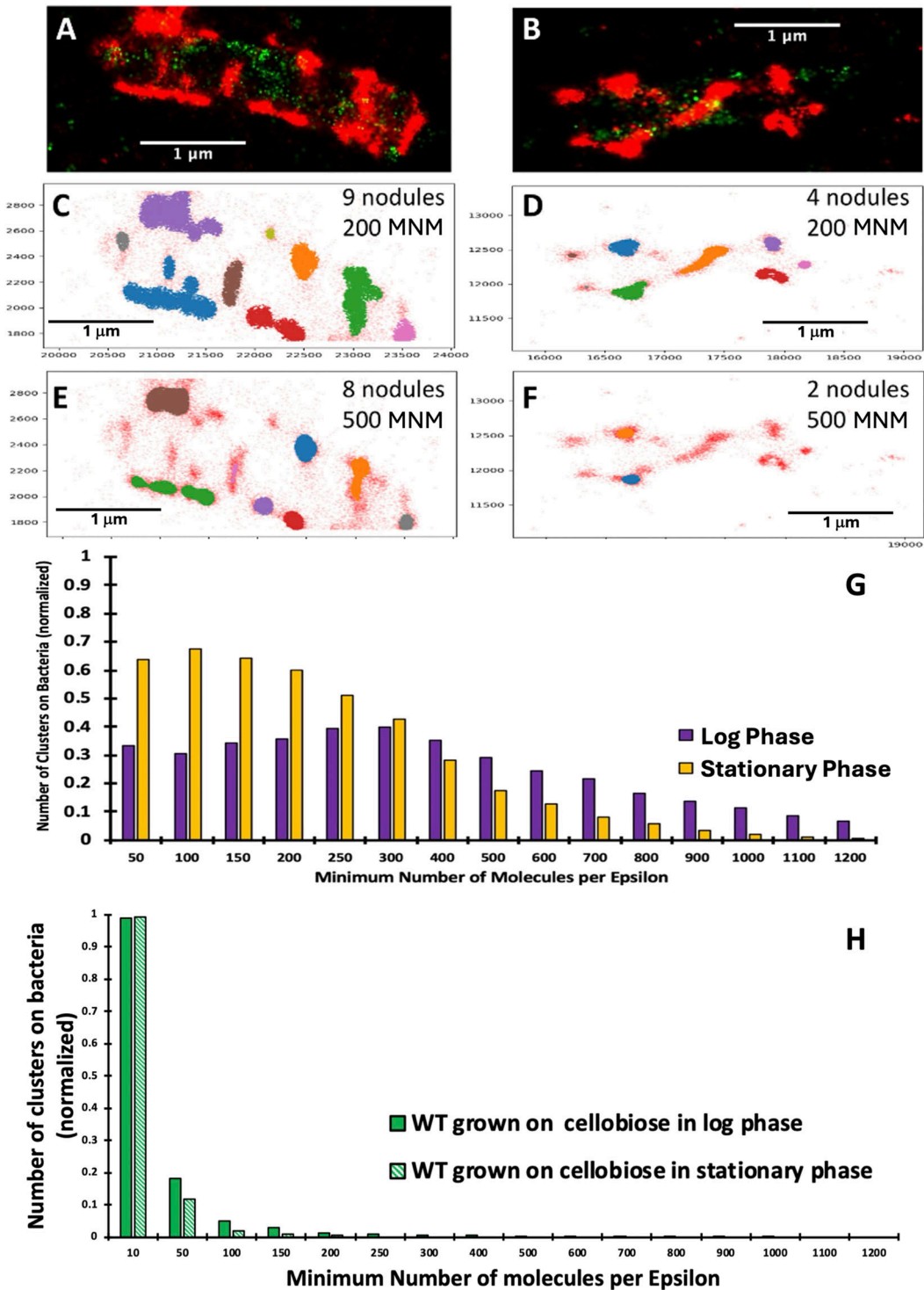

**Figure 6. *C. thermocellum* grown on cellobiose in exponential growth phase.**
**(A, B)** *C. thermocellum* grown in the presence of cellobiose in the exponential growth phase (A) with an increase in the number of cellulosomes decorating the bacterial cell wall, in contrast to bacterial cells captured in the stationary phase (B) wherein the CipA scaffoldins are still present but with significantly reduced abundance. Red fluorescence denotes CBM3a (CipA scaffoldin)-associated label, whereas green fluorescence indicates unoccupied type I cohesin–related sites (see the legend to Fig 2 and text for more detail). **(A, B, C, D, E, F)** DBSCAN results from *C. thermocellum* CBM3a-tagged cellulosomes, demonstrating the location and number of clusters for individual bacteria in the log phase (A) and stationary phase (B), with the number of detected clusters for 200 minimum number of molecules (MNM) per epsilon radius (C, D) and 500 MNM per epsilon radius (E, F). **(G)** Cluster analysis of the cellulosomal protuberances on *C. thermocellum* in the log phase (purple) and stationary phase (yellow), normalized to 1. **(H)** Cluster analysis of the unoccupied cohesins on *C. thermocellum* in the log phase (dark green) and stationary phase (light green), normalized to 1. The optical images are representative images from our data.

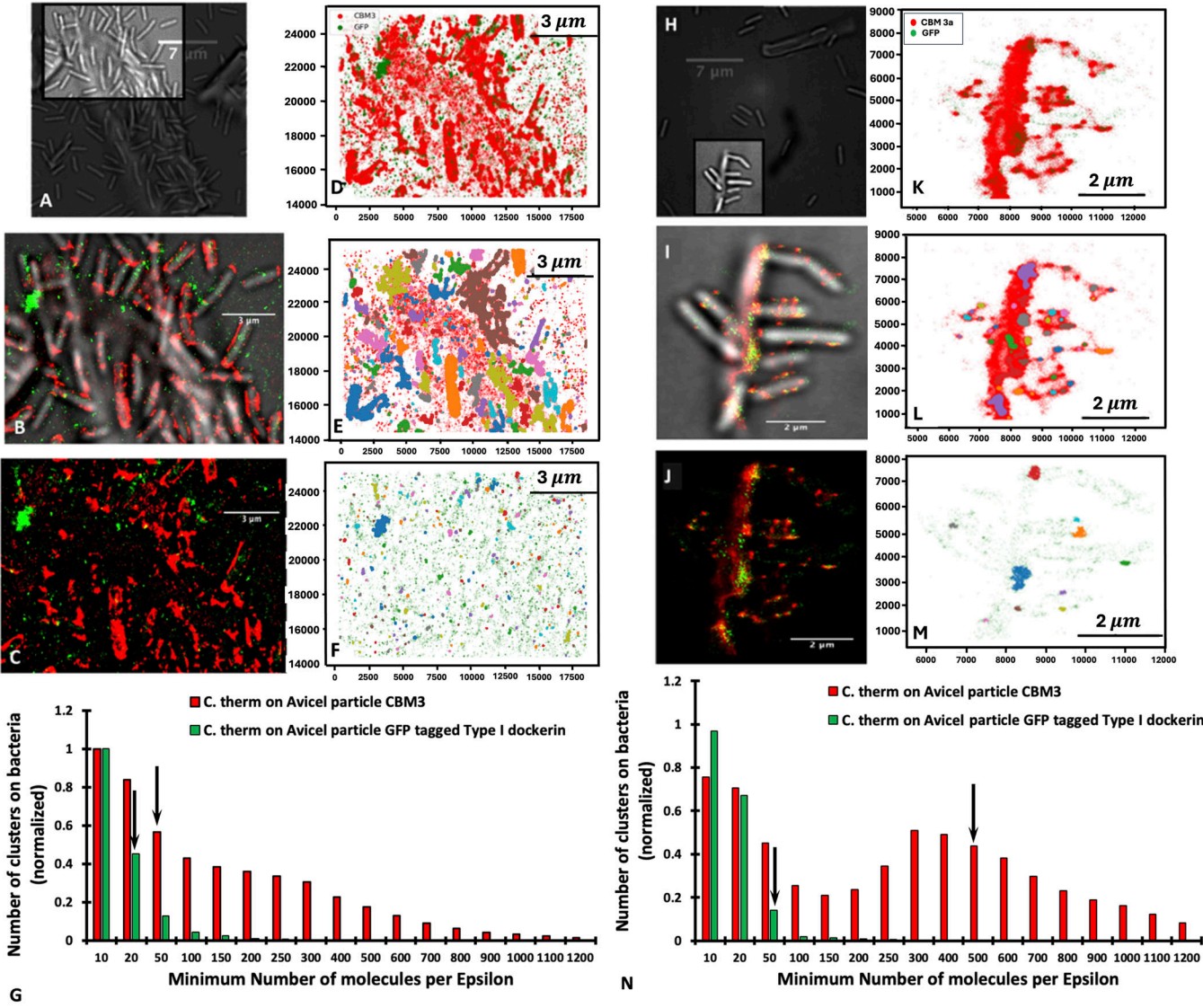

**Figure 7. Cluster plot of *C. thermocellum* grown on Avicel in log phase.**
**(A, B, C, D, E, F, G, H, I, J, K, L, M, N)** DBSCAN cluster plot of *C. thermocellum* grown on Avicel in the log phase (A, B, C, D, E, F, G) and the stationary phase (H, I, J, K, L, M, N), showing (A, H) the white-light image of the area analyzed by the clustering algorithm; ((B, I) [Fig 4A]) the overlay of the white-light and fluorescence image; ((C, J) [Fig 4B]) the fluorescence image alone; (D, K) the combination of the x-y positions for both the Alexa Fluor 647 and the photoactivatable GFP (individual plots for red and green fluorescence can be found in supplemental figure X); (E, L) the clusters identified for the Alexa Fluor 647–tagged CBM3a with an epsilon radius of 75 nm and a minimum number of molecules (MNM) set to 40 (E) or 500 (L); (F, M) the clusters identified for the photoactivatable GFP fusion protein with an epsilon radius of 125 nm and a minimum number of molecules of 20 (F) or 50 (M); and (G, N) the number of clusters versus the minimum number of points for CBM3a (red) and for GFP-dockerin (green). The arrows within the bar graph represent the minimum number of points chosen for the visual representation in (E, L) (red) and (F, M) (green). Panels (I, J) also appear in Fig 4.

cellulosome, and for this dataset, the clustering algorithm identified a total of 85 clusters that were primarily located on the bacterial cell wall. Fig 7G shows the distribution of the normalized number of clusters versus the MNM/cluster. Here, we can see that for the PA-GFP tagged to the type I dockerin, the distribution appears to be exponential, whereas the decrease in the distribution of CBM3a appears to be more attenuated. At this stage of growth, there appears to be no significant amount of CipA scaffoldins on the Avicel surface, compared with the bacterial surface, as indicated in Fig 7E. These samples collected midway through the

log phase suggest that most of the CipA scaffoldins are attached to the bacterial cell, similar to that shown for the cellobiose-grown *C. thermocellum* during the log phase.

When analyzing the bacteria that are attached to Avicel in the stationary phase, there is a significant degree of colocalization of CipA-CBM3a and the type I cohesin, primarily located on the Avicel particle at the point where the bacteria are attached to the Avicel particle (Fig 7H–L). In addition, the Avicel particle seems saturated with free cellulosomes at this stage. Fig 7L and M highlights clusters from the CipA scaffoldin molecules with the epsilon radius set to

75 nm with the MNM per epsilon radius set to 500 and clusters from the PA-GFP tagged to the type I dockerin with the epsilon radius set to 125 nm with an MNM set to 50, respectively. This clustering analysis demonstrates two concepts: (1) there is a higher concentration of CipA scaffoldin molecules within 75 nm on the Avicel in comparison with the bacteria (as shown with the shaded region) and (2) the type I dockerin clustering is primarily located on the Avicel particle. The analysis of these bacteria, shown in Fig 7N, reveals the distribution of the number of clusters on the bacteria versus the MNM per Epsilon radius, which appears to be slightly left skewed (longer on the left side of its peak than on its right). These results suggest that during the stationary phase, there is a higher concentration of CipA scaffoldin molecules that are tightly packed within the defined epsilon radius on the surface of the Avicel substrate. This is different compared with the concentration of CipA scaffoldins in the log phase. This demonstrates that during the life cycle of the bacterium, cellulosomes are gradually detached from the bacterial cell wall and adhere to the surface of the Avicel. This is exemplified by the center of the distribution being shifted toward a higher MNM per Epsilon radius and fewer overall numbers of clusters.

### *C. thermocellum* retains more cellulosomes during the log phase when detached from insoluble substrate as compared to bacteria attached to the substrate

The analysis of substrate-free bacteria revealed a remarkable distribution of CipA scaffoldins. The cellulosomes cover the surface of the bacterial cell. Using our clustering algorithm, we can identify 14 clusters for the CipA-CBM3a from Fig 8A that contain at least 200 fluorescent molecules per Epsilon radius (Fig 8B) but only eight clusters for the PA-GFP containing at least 10 fluorescent molecules per Epsilon radius (Fig 8C). Fig 8D shows the normalized distribution of clusters for both the CBM3a (red) and the PA-GFP (green), averaged over 69 bacterial cells. The PA-GFP shows a very similar exponential distribution to that in Fig 8G, demonstrating that the combination of unoccupied OlpA/OlpC/CipA sites is similar for detached and attached bacteria. However, the distribution of the CipA-CBM3a clusters for the detached bacteria is quite different from the ensemble distribution shown in Fig 8G, indicating more of a mixed population in the culture. In the stationary phase, our analysis of detached bacterial cells revealed a different pattern than that observed for log-phase cells. The overall intensity and distribution pattern are significantly different and show a loss in the number of cellulosomes attached to the surface of stationary-phase bacteria (Fig 8E). Fig 8F shows the representative distribution for the MNM of 20 for CipA-CBM3a. Fig 8G shows representative distributions for an MNM of 10 for the PA-GFP with 17 clusters detected for CipA-CBM3a and only seven clusters detected for PA-GFP (Fig 8H).

Considering the above data, we can now systematically compare the differences in distribution between the cellulosome and the unoccupied cohesins during different stages of growth for substrate-bound and substrate-detached bacterial cells. Fig 9A shows a clear shift in the distribution of cellulosomes on substrate-bound cells where significantly larger clusters are observed in the stationary phase. As illustrated in Fig 7I, most of these

large clusters are found at the EMS interface between the microbes and the biomass, accompanied by a large increase in fluorescence. Fig 9B, however, shows a different phenomenon for detached bacterial cells with cells that seem depleted in cellulosomes that are in smaller clusters. The comparison between bound and detached log-phase bacterial cells shows that they have a very similar distribution of cellulosomes (Fig 9C). These results are illustrated in Fig 5B wherein microbes are bound to the biomass but do not seem to have relocated their cellulosomes to the contact point with biomass as is the case in the stationary phase. However, the difference between bound and free cells in the stationary phase is striking with very large clusters on bound cells but much smaller cellulosome clusters on free cells (Fig 9D). In contrast, the distribution of the unoccupied type I cohesins appears to be unchanged under all conditions on bound and detached cells as shown in Fig 9E–H, indicating that the type I cohesin vacancies do not change during the different stages of growth. This is an important result, as it has been hypothesized that some of the smaller scaffoldins (i.e., OlpA and OlpC) could serve as a shuttle to populate cellulosomes (32). These results would seem to contradict this hypothesis as the number of unoccupied cohesins remains constant throughout growth.

## Discussion

Biomass deconstruction by thermophilic microbes is a complex and heterogeneous process. This process is even more challenging to understand because of the intricate multienzyme complex used by these cellulosomal microbes to deconstruct cellulosic substrates.

This super-resolution imaging and DBSCAN clustering analysis revealed that each cellulosome complex contains densely packed CipA scaffoldins, corresponding to tightly organized macromolecular assemblies. When grown on cellobiose, *C. thermocellum* cells in the log phase displayed more numerous and larger cellulosome clusters on their surfaces compared with the stationary phase, with CipA scaffoldins largely saturated with enzymes and few unoccupied cohesins. On insoluble Avicel, log-phase cells retained cellulosomes primarily on the cell surface, whereas stationary-phase cells relocated large clusters to the EMS interface, saturating the biomass surface. Detached cells in the stationary phase showed marked depletion of surface cellulosomes compared with bound cells, but unoccupied type I cohesin distribution remained constant across growth phases and conditions. These results indicate dynamic redistribution of cellulosomes during growth, with a functional shift toward substrate-associated degradation in the stationary phase, whereas cohesin vacancies remain largely unaffected.

Over the years, a considerable progress has been made toward understanding and exploiting cellulosomal systems to enable efficient deconstruction of cellulosic biomass. In early work to identify the *C. thermocellum* cellulosome, Bayer and Lamed observed nodulous protuberances on the surface of the microbe using transmission electron microscopy and immunolabeling (33). Using super-resolution techniques with sample preparation that

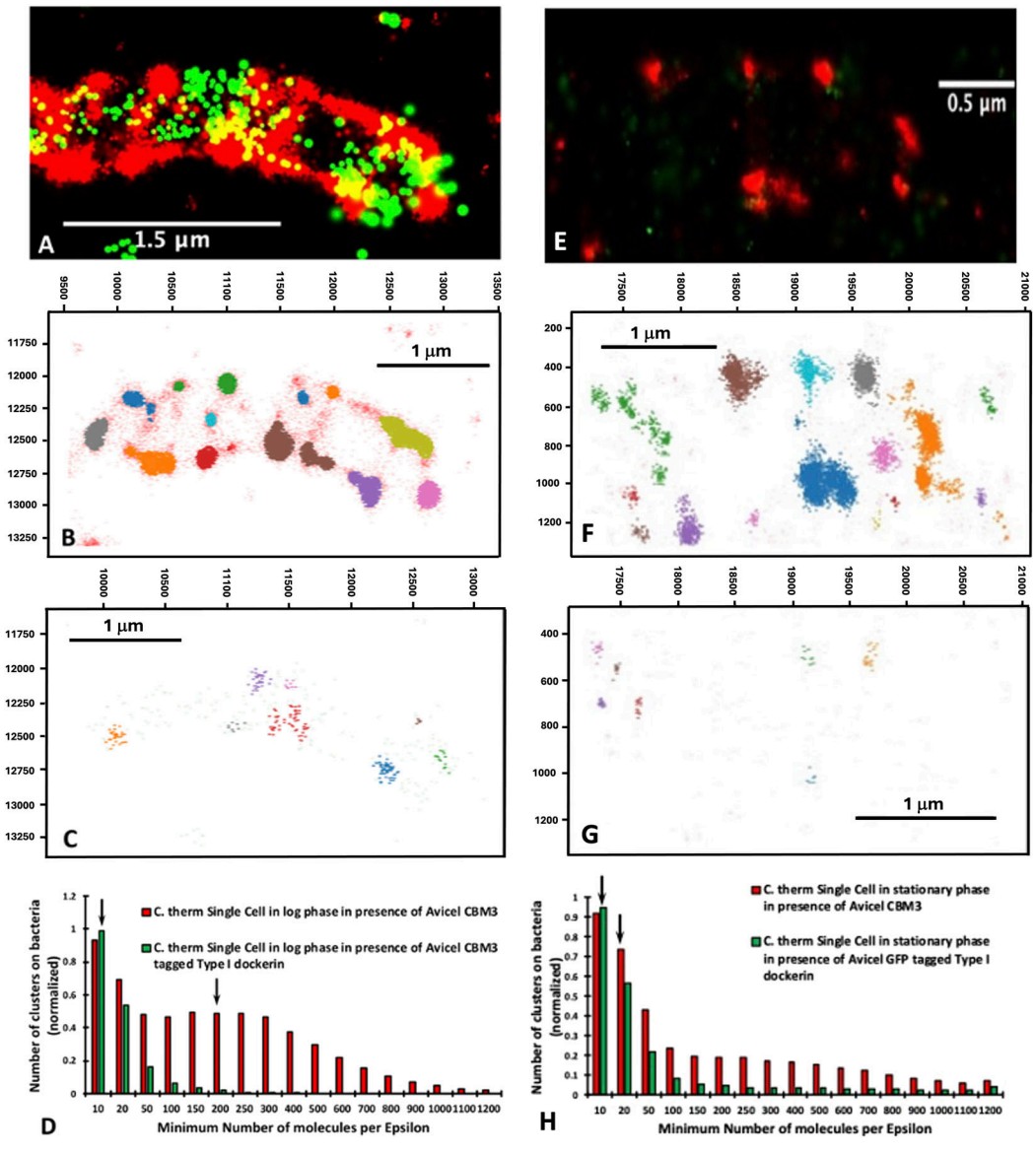

**Figure 8. Distribution of detached *C. thermocellum* cells grown on Avicel during log and stationary phase.**
**(A, B, C, D, E, F, G, H)** DBSCAN results from a detached *C. thermocellum* cell, grown on Avicel during the log (A, B, C, D) and stationary (E, F, G, H) phases of growth, showing (A, E) the original fluorescence image; (B, F) the clusters identified for the Alexa Fluor 647–tagged CBM3a with an epsilon radius of 75 nm and an MNM set to 200 (B) or 20; (C, F, G) the clusters identified for the fluorescent molecule of the photoactivated GFP fusion protein with an epsilon radius of 125 nm and an MNM of 10; and (D) the number of clusters versus the MNM for the CBM3a (red bars) and for the PA-GFP (green bars). **(B, C, F, G)** Arrows within the bar graph represent the MNM chosen for the visual representation in (B, F, C, G) for a detached bacterial cell.

more closely preserves native conditions, we observed similar protuberances in the present work (Fig 2). However, the distribution of these protuberances was different in cells grown to the log and stationary phases, when the microbe was degrading the cellulosic biomass. This finding agrees with a recent study that revealed a link between the stages of growth and the concentration of sugars and depletion of cellulosomes from the cell surface of cells grown on an insoluble cellulose substrate. Similar to the previous work by reference 28, we demonstrated herein that the microbes shed cellulosomes over the course of growth. We made the same observation when *C. thermocellum* was grown on

both soluble and insoluble substrates, which seems to indicate that the loss of cellulosomes is not triggered by their attachment to insoluble substrates. It is important to note that in the present study, these conclusions were drawn from average fluorescence values, corresponding to cellulosomes from tens of bacteria grown on different substrates, stages of growth, and those that either are or are not attached to biomass. Even though it appears that cells shed cellulosomes during growth on either soluble or insoluble substrates, it is clear from Figs 4, 7, and 9 that these cellulosomes are targeted to the biomass at the contact points in a concerted manner when substrate is present. Indeed, log-phase bound and

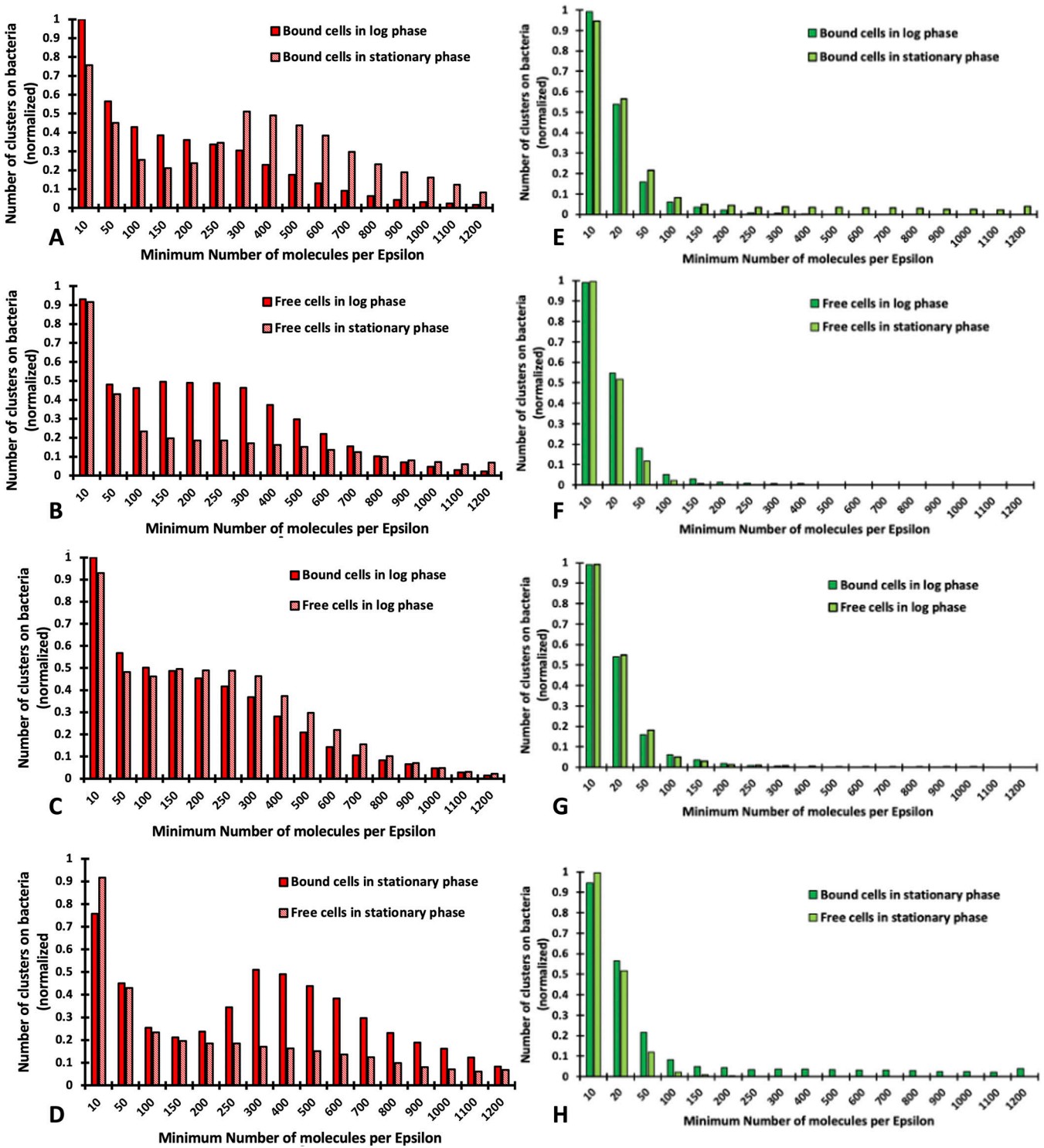

**Figure 9.    Distribution of cellulosomes for *C. thermocellum* cells in different conditions.**
**(A, B, C, D, E, F, G, H)** DBSCAN results focusing on the distribution of cellulosomes for *C. thermocellum* cells grown on Avicel, probing the effect of the same type of cells (bound or detached) being in the log or stationary phase ((A, B), respectively, for bound or detached) and probing the effect of cells being bound or detached in each stage of growth (log phase (C) and stationary phase (D)). Similar results focusing on the distribution of unoccupied type I dockerin are shown for *C. thermocellum* cells grown on Avicel, probing the effect of the same type of cells (bound or detached) being in the log or stationary phase ((E, F), respectively, for bound or detached) and probing the effect of cells being bound or detached in each stage of growth (log phase (G) and stationary phase (H)). These analyses show little differences in the distribution of unoccupied type I dockerins on the *C. thermocellum* cells in the different growth stages.

detached cells exhibit the same distribution profiles for the number of molecules per cellulosome cluster. However, in the stationary phase, detached bacterial cells bear much smaller cellulosome clusters, whereas bound cells exhibit much larger cellulosome clusters (Fig 9). We showed that stationary-phase bound cells in contact with the cellulosic biomass seem to reposition and concentrate the cellulosomes at the contact points or EMS interface (Fig 4). These results are consistent with early TEM observations (3, 15) in which the exocellular protuberances of cellulose-bound cells were reported to undergo dramatic conformational change to form contact corridors where the cellulosomes are concentrated on the surface of the cellulosic substrate. In addition, there is an increased colocalization of empty type I cohesin and cellulosomes at these contact points (Fig 4), indicating that some of the dockerin-bearing enzyme molecules occupying these cohesin-bearing scaffoldins were released to interact directly with the substrate. The unoccupied type I cohesin can be found on different protein scaffoldins, for example, OlpA and OlpC, attached to the microbial cell wall. These structures are also found on the primary scaffoldin, CipA, and can be either detached or initially bound to the bacterial cell wall via the secondary scaffoldin OlpB (Fig 1). In Fig 4, it is indeed possible to differentiate between the empty type I cohesin on the microbe (OlpA and OlpC) and that on the detached cellulosomes (CipA). It is also expected that unoccupied type I cohesins on CipA would be found close to the CipA-CBM3a and colocalized. From our data, it appears that most of the unoccupied type I cohesins are located on the microbes and should correspond to OlpA and OlpC, given that they are not in the vicinity of CipA-CBM3a, especially on detached cells or in the log phase. It is also interesting to note that the size of the clusters associated with the unoccupied type I cohesin does not change over the course of growth or when the bacteria are grown on different substrates. OlpA and OlpC have been hypothesized to be involved in shuttling enzymes to cellulosomes (32), but from our data, there is no indication that this would be the case. This conclusion is further reinforced, given the fact that the abundance of unoccupied type I cohesins does not change over time, at least within the constraints of our experiments.

One notable phenomenon that occurs in the stationary phase of growth is the saturation of substrate with cellulosomes. Indeed, the bacterial cells appear to shed cellulosomes over time, and they are clearly being shuttled to the biomass leaving very few unoccupied spots on the surface of the substrate. This phenomenon is consistent with early TEM studies (3, 15) and could explain some of the slowdowns during the growth of *C. thermocellum* on insoluble substrates, whereby the saturation of the substrates with cellulosomes could be counterproductive and lead to "traffic jams" of the cellulases on the substrate. In addition, in all cases, the concentration of cellulosomes, whether on the microbe during the log phase or on the substrate during the stationary phase, is much higher than anticipated.

This study highlights the potential of using these new optical techniques, combined with specific mutations in *C. thermocellum* or other cellulosomal microbes, to probe and challenge other hypotheses that have been proposed regarding cellulosomes and their mode of action, such as the role of "shuttle" scaffoldins and the attachment of *C. thermocellum* to biomass. These advanced

imaging tools will also serve to optimize the process of cellulose deconstruction by these microbes in pure cultures or in cocultures. We note that for cellulases, more is not necessarily better in industrial settings and that controlling cellulosome expression levels could ultimately be important for better biomass deconstruction in these settings.

# Materials and Methods

### Experimental method

*C. thermocellum* cultures were grown in serum bottles containing MTC medium supplemented with either cellobiose or Avicel with each serum bottle dedicated to a single biological replicate. Samples were harvested during both logarithmic and stationary growth phases.

For imaging preparation, 0.5 ml aliquots were withdrawn and washed in 50 mM sodium acetate buffer containing 2 mM CaCl$_2$ (sample preparation buffer) to remove residual medium components and free cellulosomes. These washed cells were incubated for 10 min with 0.75 $\mu$g/$\mu$l CBM3a antibody, 0.387 $\mu$g/$\mu$l Alexa Fluor 647, and 0.387 $\mu$g/$\mu$l photoactivatable GFP (PA-GFP). After labeling, samples were washed three times with the same sample preparation buffer. Labeled cells were then immobilized onto (specify coverslip type, e.g., No. 1.5H high-precision) coverslips.

For STORM imaging, the imaging buffer consisted of PBS supplemented with 0.5 mg/ml glucose oxidase, 100 mM $\beta$-mercaptoethylamine (MEA), and ~10 $\mu$l potassium chloride, prepared fresh before imaging to ensure optimal photoswitching conditions.

### Imaging

For each sample coverslip, fresh STORM/PALM imaging buffer was prepared using Tris buffer, cysteamine hydrochloride (MEA), and glucose oxidase (GLOX). The pH value was adjusted to 8 using KOH. The buffer was then filtered through a 0.2-$\mu$m syringe filter and applied directly onto the sample coverslip, which was mounted in an imaging chamber (Invitrogen Attofluor Cell Chamber). A second coverslip was placed on the top to block CO$_2$ uptake. The STORM/PALM data were taken on a Zeiss Elyra P1 microscope equipped with four excitation lasers (405 nm/50 mW, 488 nm/200 mW, 561 nm/150 mW, and 642 nm/200 mW) and an Andor iXon 897 EMCCD camera with a 512 × 512 back-illuminated chip. A Zeiss Alpha Plan-Apochromat TIRF 100×/1.46 NA oil objective was used for imaging, which yields a pixel size of 100 nm when imaging with the additional 1.6× magnification tube lens. Samples were illuminated in ultra-high-power HiLo mode, which illuminates a usable area of 256 × 256 pixels (25.6 × 25.6 $\mu$m). Fluorescence images were taken with separate filter cubes for single-channel fluorescent imaging, which, together with a laser blocking filter in front of the camera, resulted in emission detection band passes of 655–725 nm for the AF647 and 500–550 nm for the PA-GFP channel, respectively. Robust blinking in the STORM/PALM imaging channels was achieved using 100% relative excitation power, and the

density of the blinking fluorophores during the experiment was controlled by an additional 405-nm laser illumination at relative powers up to 1%. 15,000 were acquired in time-series acquisitions for each STORM/PALM channel. The resulting time series were first processed using temporal median filtering (34), then analyzed with the ThunderSTORM plugin for ImageJ (35). Fluorophore localizations were corrected for sample drift (36), filtered for localization accuracy, and visualized accordingly.

## Data analysis

Super-resolution techniques such as PALM and STORM incorporate single-molecule identification and generate large datasets consisting of the molecular coordinates (x-y location) of each molecule detected. Modern-day algorithms exist to identify and locate these molecules but lack the ability to characterize any potential pattern generated by the molecules. Here, we use unsupervised machine learning (also referred to as cluster analysis) to determine whether there are any patterns formed with the distribution of cellulosomes on the bacterial cell wall and on cellulose. Unlike supervised machine learning, unsupervised learning does not use a training set, but instead finds previously unknown patterns within the dataset without preexisting labels, and is widely used in clustering, dimension reduction, and data representations (37, 38). Within the field of unsupervised machine learning, there exist multiple techniques ranging from hierarchical clustering, k-means, density-based spatial clustering of applications with noise (DBSCAN), and ordering points to identify the clustering structures (OPTICS) and others.

Cluster analysis has been used within the context of super-resolution techniques, as described by the authors in reference 39, clustering, that is, grouping the objects of a database (in this case, the single molecules), into meaningful subclasses. In this work, we decided to use DBSCAN cluster analysis, developed originally by the authors in reference 40. As defined by these authors, the key idea behind the DBSCAN algorithm is that for each point of a cluster, the neighborhood of a given radius has to contain at least a minimum number of points; that is, the density in the neighborhood has to exceed some threshold (40). In addition, the shape of the neighborhood is determined by the choice of a distance function for two points (40). Therefore, DBSCAN only requires two parameters, the radius of a defined area and the MNM within that defined area, which allows the discovery of clusters of arbitrary shape. We used DBSCAN from the scikit-learn open-source machine-learning library for the Python programming languages (41). Scikit-learn supports supervised and unsupervised learning and provides various tools for model fitting, data preprocessing, model selection and evaluation, and many other utilities (41).

## Image processing

Multiple software packages were used in the analysis of these datasets, including MATLAB for the processing of the PALM/STORM raw data, ImageJ/Fiji (38), and python with the following packages: pandas, openCV, pillow, numpy, and matplotlib. Fiji was primarily used to view the processed images generated from the MATLAB code, as well as the white-light images. Fiji was also used to

perform image processing, including resizing of the image and merging the different fluorescent images with the white-light image. Once the images were processed, they were fed into the python script, written to allow us to select regions of interest (ROI), and capture the ROI from the .csv files, containing the coordinates of each biomolecule. These scripts allowed us to only perform cluster analysis within the ROI, thus saving computational time and reducing the influence of noise. After the selection of the ROI, the x-y coordinates from each of the biomolecules were then analyzed using python and the scikit-learn-DBSCAN algorithm was used to characterize their clustering.

## Fluorescent probes

Three fluorescent probes were chosen for the project, based upon the wavelengths available with the Zeiss Elyra super-resolution imaging. Those chosen were Alexa Fluor 647 (23, 25) and PA-GFP (30). Alexa Fluor 647 was chosen because of its high blinking efficiency and overall use for STORM imaging. PA-GFP was chosen because regular GFP had been expressed with a dockerin domain that was associated with the type I dockerin. Therefore, we employed the mutation by switching the histidine with the tyrosine (30). This mutation was performed in an *Escherichia coli* expression system, originally designed for the type I dockerin (42).

## Primary antibody expression

Polyclonal antibodies against the CBM3a of the CipA scaffoldin were elicited in rabbits and purified according to the procedure described in reference 43.

## GFP expression

To express the modified GFP, we used a plasmid using Q5 High-Fidelity DNA Polymerase (New England BioLabs) according to the manufacturer's instructions. The codon-optimized CtDoc-GFP expression vector was synthesized by GenScript. pDCYB 209, containing the point mutation (T203H) at amino acid position 203 of the GFP coding sequence as shown in Fig S1, was generated using overlapping PCR (44). A 6.514-kb DNA fragment was synthesized with the forward primer DCB619 (ACCTGTCGCATCAATCTGCCCTTTCGA AAGATCCCAACGA) and reverse primer DCB620 (AGGGCAGATTGATGC GACAGGTAATGGTTGTCTGGTAAAAGGAC) using the CtDoc-GFP expression vector as a template. *E. coli* strain DH5a cells were transformed by electroporation in a 2-mm path-length cuvette at 2.5 V, and transformants were selected for kanamycin resistance. The sequences of pDCYB 209 were confirmed by automatic sequencing (Genewiz).

*E. coli* BL21(DE3) was used for protein expression. Recombinant strains were grown in LB broth supplemented with kanamycin (25 μg/ml). Cultures were induced at 15°C with 0.2 mM IPTG when $OD_{600}$ = 0.4. Cultures were centrifuged at 5,000$g$ for 10 min when OD600 ≥ 1.2. To recover intracellular protein, centrifuged cells were enzymatically lysed in Buffer A (50 mM Tris–HCl, 40 mM NaCl, 10 mM imidazole, pH 8.0) supplemented with lysozyme, protease inhibitors, and nuclease (Pierce) at 4°C. Enzymatically lysed cells were subjected to 1 min of sonication in a water bath at 10-s intervals

punctuated by 30 s on ice. Lysed cells were centrifuged at 10,000$g$ for 30 min. The lysate was further purified via immobilized metal affinity chromatography purification using a 5-ml HisTrap FF Crude column (GE Healthcare). Once bound, Buffer B (Buffer A with 200 mM imidazole) was used to elute the protein. The recovered fractions were then purified further via size-exclusion chromatography using a HiLoad 16/600 Superdex 200 prep grade column (GE Healthcare). This step served to buffer-exchange the fractions into SEC buffer (200 mM acetate, 100 mM NaCl, 10 mM CaCl$_2$, pH 5.5). Once collected, the recovered protein was concentrated using a 10-kD spin concentrator (Sartorius). The final protein concentration was determined using a Pierce BCA protein assay (Pierce). The purified CtDoc-GFP (PA) protein was analyzed by SDS–PAGE using 4–12% NuPAGE Bis-Tris Gel (Invitrogen) run at 150 V for 55 min in MOPS SDS buffer. The gel was then stained with Colloidal Blue (Invitrogen).

### Bacterial growth and characterization

Growth studies were performed with DSM1313 and CTN5 strains in 50 ml total volume in anaerobic serum bottles, containing 20 ml of CTFUD media with 2% D(+)-cellobiose (ACROS Organics) or 0.5% Avicel PH-101 (Sigma-Aldrich) as the carbon sources. Seed cultures were grown at 60°C from frozen stock in 20 ml of culture medium, containing 0.5% cellobiose CTFUD media. Upon reaching the exponential phase, samples were transferred from seed bottles to fresh bottles containing either 2% cellobiose or 0.5% Avicel, where the initial optical density (OD) was set at ~0.045. Transferred cultures were grown at 52°C with agitation (~0.14$g$ for cellobiose bottles, ~0.57$g$ for Avicel bottles). The cultures were grown overnight for 5–8 h, after which samples were taken periodically until the cultures reached the steady state.

OD sampling was performed using a Genesys 20 spectrophotometer (Thermo Fisher Scientific) at 600 nm. When sampling Avicel bottles, 1.5-ml samples were centrifuged at 0.3 min$^{-1}$ × g for 30 s to precipitate the Avicel. The supernatant fluids were sampled using a pipette and used to determine the sample OD.

### Chromophore and GFP attachment

For the two chromophores, two primary antibodies were expressed, one primary antibody expressed in rabbit for the CBM3a located on the CipA scaffoldin and another primary antibody expressed in rabbit that is associated with the GH48 enzyme. WT or the mutant bacterial cells were grown with several passages during the log phase to ensure a healthy culture. Once the bacterial cells were in the appropriate growth stage (either the log phase or the stationary phase), 2 ml of the bacterial cultures was extracted from the serum bottles, centrifuged, and washed three times using 50 mM SEC buffer supplemented with 5 mM CaCl. The cells were then resuspended in the SEC buffer, both primary antibodies were added, and the suspension was incubated at room temperature for 5 min. The immunolabeled bacterial cells were washed an additional three times with the SEC buffer by centrifugation as described above. The primary antibody-tagged bacterial cells were then incubated with the secondary antibody tagged with AF647 along with the PA-GFP for 5 min. The bacterial cells were washed by centrifugation a final three times in PBS buffer and stored in PBS buffer to match the imaging buffer used in the experiments. Sample preparation and imaging occurred on the same day. Labeled bacterial cells were drop-cast onto a poly-L-lysine–coated 25-mm round glass coverslips.

## Data Availability

The datasets generated and analyzed during the current study are available from the corresponding author upon reasonable request. Because of the very large file sizes associated with PALM/STORM raw image stacks and derived quantitative analysis files, the complete datasets exceed the upload limits of commonly used public repositories (e.g., GitHub and standard data-hosting services). Requests for access to the data can be directed to YJ Bomble (Yannick.Bomble@nrel.gov). Data will be shared for noncommercial research purposes in accordance with institutional data-sharing policies.

## Supplementary Information

## Acknowledgements

This material is based upon work supported by the Center for Bioenergy Innovation (CBI), U.S. Department of Energy, Office of Science, Biological and Environmental Research Program under Award Number ERKP886. This work was authored in part by Alliance for Sustainable Energy, LLC, the Manager and Operator of the National Renewable Energy Laboratory for the U.S. Department of Energy (DOE), under Contract No. DE-AC36-08GO28308. The views expressed in the article do not necessarily represent the views of the DOE or the U.S. Government. The U.S. Government retains and the publisher, by accepting the article for publication, acknowledges that the U.S. Government retains a nonexclusive, paid-up, irrevocable, worldwide license to publish or reproduce the published form of this work, or allow others to do so, for U.S. Government purposes.

### Author Contributions

JM Yarbrough: conceptualization, data curation, software, formal analysis, visualization, methodology, and writing—original draft.
NN Hengge: investigation and writing—review and editing.
Q Xu: investigation and writing—review and editing.
SJ Ziegler: data curation, formal analysis, and writing—review and editing.
D Chung: investigation and writing—review and editing.
S Huang: investigation and writing—review and editing.
S Moraïs: formal analysis and writing—review and editing.
I Mizrahi: formal analysis and writing—review and editing.
EA Bayer: formal analysis and writing—review and editing.
YJ Bomble: conceptualization, resources, supervision, funding acquisition, validation, project administration, and writing—original draft.

## Conflict of Interest Statement

The authors declare that they have no conflict of interest.

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
