## [Reviewer comments · Life Science Alliance]

Deconstruction by *C. thermocellum* - From Microbe Mediated to Dynamic Redistribution of Cellulosomes

John Yarbrough, Neal Hengge, Qi Xu, Daehwan Chung, Samantha Ziegler, shu Huang, Sarah Moraïs, Itzhak Mizrahi, Edward Bayer, and Yannick Bomble

DOI: <https://doi.org/10.26508/lsa.202503239>

Corresponding author(s): Yannick Bomble, National Renewable Energy Laboratory

Review Timeline:

Submission Date:	2025-01-27
Editorial Decision:	2025-03-13
Revision Received:	2025-08-27
Editorial Decision:	2025-09-25
Revision Received:	2025-12-09
Accepted:	2025-12-15

Scientific Editor: Tim Fessenden

Transaction Report:

March 13, 2025

Re: Life Science Alliance manuscript #LSA-2025-03239-T

Dr. Yannick Bomble
NREL
15013 Denver W Pkwy
Golden, CO 80401

Dear Dr. Bomble,

Thank you for submitting your manuscript entitled "Understanding the Dynamics of Biomass Deconstruction by the Cellulolytic Anaerobe *C. thermocellum*" to Life Science Alliance. The manuscript was assessed by expert reviewers, whose comments are appended to this letter. We invite you to submit a revised manuscript addressing the Reviewer comments.

Thank you for this interesting contribution to Life Science Alliance. We are looking forward to receiving your revised manuscript.

Sincerely,

B. MANUSCRIPT ORGANIZATION AND FORMATTING:

Reviewer #1 (Comments to the Authors (Required)):

Yarbrough et al provide a manuscript on super resolution fluorescence microscopy imaging of cellulosomes on the surface of *Clostridium thermocellum* cells. They show that the degree of surface coverage of cellulosomes changes under different growth conditions, and dependent on substrate contact. It is also suggested that dockerin binding sites on CipA are mostly occupied while they are not on other modules.

Overall, the manuscript is quite diffuse, as is the title. In the abstract, it is stated "providing a clearer picture of the dynamics of cellulosome populations at the enzyme microbe substrate interface." In what way do the authors think the view is now clearer? In our view, the ms would gain much if the message was sharpened, as to what new insights were obtained.

Main comments

- 1) I am unsure about the composition of cellulosomes, and the description in the text. Fig. 1: what is the relation between the CipA molecule on the left and the structures embraced by the bracket: does the cell have different types of CipAs bound to different "Olp" proteins within the membrane (or is it the cell wall)? A more precise cartoon would be very helpful.
- 2) Why is the green signal from PA-GFP type-I dockerin fusion protein mostly separate/distinct from the antibody-detected red fluorescence for CipA? CipA contains 9 dockerin domains, so why aren't these detected? The authors state "Surprisingly, it seems that most empty type-I cohesins are located on the bacterial cell wall and that the CipA scaffoldins are almost fully saturated with enzymes." That seems very unlikely, except if the Olp dockerin domains have an enormously higher dissociation constant (i.e. turnover) than those on the CipA protein. Is there evidence for this? If not, the authors should do a pulse chase, in which they add the PA-GFP dockerin fusion and wash it off, and do imaging without fixation (please see below - not clear how any imaging experiment was done).
- 3) Fig. 2A - it seems obvious where the outlines of the cells are, but not completely clear. Please draw outlines with an e.g. thin white line, or make a cartoon underneath the FM image, if there is no appropriate bright field image available for the image.
- 4) The manuscript does not even contain page numbers, let alone line numbers. That's quite a pain for any reviewer. Section II, 1. "The main drawback of traditional optical microscopy is the inability to systematically quantify the information within the image. Unfortunately, with optical images, it is difficult to quantify the fluorescence from each bacterium, let alone analyze any significant portion of the bacterial cells." Why would that be? There are plenty of programs out there to automatically quantify fluorescence of fields of cells, or within single cells ("Mircobetracker", for example). Please give the educated reader appropriate reasoning why you believe in what is stated.
- 5) As I take it, a main objective of the manuscript is to provide a protocol for super resolution imaging of cellulosomes. However, there is not a single line in the methods about imaging. Please fill in how STORM and PALM was done, what were the settings.
- 6) Text above Fig. 2 "with minimal sample preparation and disruption of the bacterium, with a resolution of 35 nm." Please show how you arrived at this resolution, and how sample preparation was done (see point 5). Please provide evidence that the red signals for the STORM signals are composed of single molecules (single blinking events?). The images don't look like super resolution (SR) images to me, except for the PA-GFP signals. Speaking of SR - I have so far not heard "super high resolution", just either super - or high resolution. It seems very peculiar that the authors would try to top the term "SR" imaging.

Below, please find mostly textual comments.

Result 1

1. "cell-free cellulosomes" could be "free cellulosomes"
 - o "Cell-free cellulosomes" is somewhat redundant because "cellulosomes" are defined as extracellular structures, and sounds like cellulosomes arose independent from cells. "Free cellulosomes" would make more sense.
2. "It was further proposed that detachment of the bacterial cells may be connected to a controlled release of the cells from the cellulose-bound cellulosome, which would continue to hydrolyze the substrate."
 - o : unclear phrasing
3. "examined at nanoscale resolution single *C. thermocellum* cells"
 - o : Incorrect word order; should be "examined single *C. thermocellum* cells at nanoscale resolution."
4. "observed phenotypic cellulosomal heterogeneity mediated by soluble sugar concentration in the media"

- o : Incorrect use of "mediated by." The soluble sugar concentration is likely influencing or regulating, not "mediating."
- 5. "suggesting a division-of-labor strategy"
 - o : "Division of labor" does not require hyphens unless used as a compound adjective.
- 6. "further clarify the status of cellulosomes with regards to the presence of both the primary scaffoldin and the cellulosomal enzymes on cells of *C. thermocellum* and on the substrate"
 - o : "With regards to" could be "with regard to"
- 7. "we used a primary antibody targeting the CBM3a of the CipA scaffoldin with a secondary antibody fused to Alexa Fluor 647"
 - o Unclear sentence. Use "and" instead of "with" or any other modification to clarify what was done
- 8. "a PA-GPF" Should be "a PA-GFP"
 - o : Typographical error ("PA-GPF" instead of "PA-GFP").
- 9. "grown on cellobiose in the stationary phase ."
 - o : Extra space before the period.
- 10. "we were able to further validate these original findings, with minimal sample preparation and disruption of the bacterium, with a resolution of 35 nm."
 - o : Too many instances of "with," making the sentence clunky.
- 11. "To date, these cellulosomal protuberances and their dynamics have not been fully characterized, especially on process-relevant feedstocks."
 - o : "Process-relevant feedstocks" is unclear. It would be better to specify what type of feedstocks (e.g., lignocellulosic biomass).
- 12. "These advanced optical microscopy techniques appear to be particularly well suited to the quantitative study of the population of cellulosomes and their dynamics over time during biomass deconstruction."
 - o : "Study of the population of cellulosomes", Better phrased as "quantitative analysis of cellulosome populations and their dynamics during biomass deconstruction."

1. "cell-free cellulosomes on the surface of cellulose"

- o : Cellulosomes are usually attached to cells or substrates. If they are truly cell-free, clarification is needed.
- 2. "Bayer and coworkers discovered these protuberances on *C. thermocellum* and reported that both their size and longitudinal arrangement varied as determined by either a selective cationic electron-dense probe or antibodies specific to the CipA scaffoldin"
 - o : The phrase "as determined by" is misleading because the size and arrangement of protuberances do not depend on the labeling technique-they were observed using these methods.

Result 2

- "From these data, there is a distinct change in the pattern of the polycellulosomal protuberances which form highly decorated and interconnected protuberances on the bacterial cells in the log phase."
 - o : Redundant phrasing ("polycellulosomal protuberances" vs. "highly decorated and interconnected protuberances"). Also, "From these data, there is" is an awkward construction.
 - "These new structures appear to be depleted when the microbe is in the stationary phase of growth."
 - o : "Appear to be" is unnecessary unless there is doubt. "Depleted" is better suited for resources rather than physical structures.
 - "Surprisingly, it seems that most empty type-I cohesins are located on the bacterial cell wall and that the CipA scaffoldins are almost fully saturated with enzymes."
 - o : "It seems that" weakens the statement.
 - "We utilized the DBSCAN cluster analysis with the number of clusters per bacterial cell versus the MNM per epsilon."
 - o : The sentence is incomplete; it does not clarify what was being analyzed.
- o Wordy Phrasing
- "Red fluorescence denotes CBM3a (CipA scaffoldin)-associated label, whereas green fluorescence indicates unoccupied type I cohesin-related sites (see legend to Figure 2 and text for more detail)."
 - o : "See legend to Figure 2" → should be "See the legend in Figure 2."
 - "We also give a visual example of the number of clusters and their overall size on a specific bacterial cell in the log phase and in the stationary phases of growth."
 - o : "We also give" is informal. "Specific bacterial cell" is redundant.
 - "For example, for 500 MNM per epsilon, two nodules were detected in the stationary phase, but eight nodules were detected in the log phase, thus illustrating that the nodules on the bacterial cells are larger in the log phase."
 - o : Repetitive use of "nodules" and "log phase."
 - "Altogether, we analyzed the normalized numbers of clusters collected from over 60 single bacterial cells in both the log and stationary phases of growth."
 - o : "Single bacterial cells" is redundant (all bacterial cells are single unless otherwise stated).
 - "The reasoning for normalizing the number of clusters per bacterial cell is to gain quantitative insight into how the density of molecules within the cellulosomal protuberances is affected during growth."
 - o : "The reasoning for" is clunky.
- "The signal for the unoccupied type-I cohesins and these analyses indicate that they are fairly dispersed on the surface and not co-localized with cellulosomes, which indicates either a low abundance of unoccupied type-I cohesins or a low abundance of type-I cohesins on the surface of these bacteria."
 - o : Confusing conclusion-does it mean unoccupied cohesins are low in number, or that they are not clustering?

In result 3: Rephrase this sentence, the idea is vaguely expressed

"When analyzing the stationary phase when attached to Avicel, we found a significant amount of colocalization of the CipA-CBM3a and the type-I cohesin, which is primarily located on the Avicel particle and in particular at the point where the bacterium is attached to the Avicel particle (Figure 7H-J)."

The paragraph is mostly well-structured, but there are a few grammatical and stylistic issues. Here are the key mistakes:

1. "When analyzing the stationary phase when attached to Avicel"

o Issue: Awkward phrasing and unclear subject. "When analyzing" implies that "we" (the researchers) are the subject, but "the stationary phase when attached to Avicel" is unclear. The stationary phase itself is not "attached"-the bacteria are.

2. "found a significant amount of colocalization"

o Issue: "Amount" is generally used for uncountable nouns, but "colocalization" is more abstract. "Significant degree of colocalization" is a better phrasing.

3. "the CipA-CBM3a and the type-I cohesin"

o Issue: The article "the" before "CipA-CBM3a" is unnecessary

4. "which is primarily located on the Avicel particle"

o Issue: The relative pronoun "which" refers to "type-I cohesin," but the sentence makes it sound like both "CipA-CBM3a and type-I cohesin" share this location. This is ambiguous.

5. "and in particular at the point where the bacterium is attached to the Avicel particle"

o Issue: "And in particular" is redundant. Also, "the bacterium" implies only one, but it seems to refer to bacteria in general.

Result 4

• "The cellulosomes appear to cover the surface of the bacterial cell."

o : "Appear to" is unnecessary unless there is uncertainty.

• "The PA-GFP shows a very similar exponential or geometric distribution to that in Figure 5G, demonstrating that the combination of unoccupied OlpA/OlpC/CipA sites is similar for detached and attached bacteria."

o : "A very similar exponential or geometric distribution" is unclear. Is it both, or one?

• "However, it appears that the distribution of the CipA-CBM3a clusters for the detached bacteria is quite different from the ensemble distribution shown in Figure 5G, which would indicate more of a mixed population in the culture."

o : "It appears that" weakens the statement. "Would indicate" is also weak.

• "The overall intensity and distribution pattern are significantly different and show a loss in the number of cellulosomes attached to the surface of stationary-phase bacteria (Figure 7E)."

o : "Loss in the number of cellulosomes" is awkward.

2. Wordy Phrasing

• "Figures 7F and G show representative distributions for an MNM of 500 for CipA-CBM3a and 50 for the PA-GFP with 17 clusters detected for CipA-CBM3a and only seven clusters detected for PA-GFP (Figure 7H)."

o : Too much information in one sentence.

Discussion

• "on either soluble and insoluble substrates"

o : Incorrect use of "either" with "and." "Either" should be followed by "or," not "and."

• "attached or attached to biomass"

o : Repetition-either "attached" was mistakenly repeated, or "detached" was intended.

• "when the microbe was actively engaged in degradation of cellulosic biomass"

o : "Engaged in degradation of" is unnatural. The verb "degrading" is more concise.

• "Additionally, there seems to be increased colocalization of empty type-I cohesin and cellulosomes at these contact points"

o : "There seems to be" is weak and redundant.

• "The unoccupied type-I cohesin can be found on different protein scaffoldins, e.g., OlpA and OlpC, attached to the microbial cell wall, but also on the primary scaffoldin, CipA, either detached or initially bound to the bacterial cell wall via the secondary scaffoldin OlpB (Figure 1)."

o : The phrase "but also" disrupts parallelism. The sentence is too long and convoluted.

2. Wordy Phrasing

• "This process is even more challenging to understand when considering cellulosomal microbes that primarily use an intricate multienzyme complex to deconstruct cellulosic substrates instead of free enzymes."

o : Wordy and slightly redundant.

• "We made the same observation when *C. thermocellum* was grown on both soluble and insoluble substrates, which seems to indicate that the loss of cellulosome is not triggered by their attachment to insoluble substrates."

o s:

1. "Loss of cellulosome" should be plural ("cellulosomes").

2. "Their" does not agree in number with "cellulosome."

• "It is also expected that unoccupied type-I cohesins on CipA would be found close to the CipA-CBM3a and are therefore considered to be colocalized."

o : "Expected" and "considered" introduce unnecessary uncertainty.

3. Redundancy & Repetition

- "Indeed, log-phase bound and detached cells appear to exhibit the same distribution profiles for the number of molecules per cellulosome cluster."
 - o : "Appear to exhibit" is wordy.
- "One notable phenomenon that occurs in the stationary phase of growth is the sheer saturation of substrate with cellulosomes."
 - o : "Sheer saturation" is redundant-saturation already implies completeness.
- "Again, this phenomenon is consistent with early TEM studies [3, 15] and could potentially explain some of the slowdowns that happen during growth of *C. thermocellum* on insoluble substrates, whereby the saturation of the substrates with cellulosomes could be counterproductive and lead to 'traffic jams' of the cellulases on the substrate."
 - o s:
 1. "Again" is unnecessary.
 2. "Could potentially" is redundant-use either "could" or "potentially."
 3. "Slowdowns that happen during growth" can be simplified.

4. Ambiguity & Logical Issues

- "These results are consistent with early TEM observations [3, 15] whereby the exocellular protuberances of cellulose-bound cells were reported to undergo dramatic conformational change to form contact corridors where the cellulosomes are concentrated on the surface of the cellulosic substrate."
 - o : "Whereby" is improperly used-use "in which."
- "This study highlights the potential of using these new optical techniques, combined with specific mutations in *C. thermocellum* or other cellulosomal microbes, to probe and challenge other hypotheses that have been proposed regarding cellulosomes and their mode of action, such as the role of 'shuttle' scaffoldins and the role of the attachment of *C. thermocellum* to biomass."
 - o : "The role of the attachment of *C. thermocellum* to biomass" is awkward.

5. Formatting Issues & Typos

- "[tal] et al [29]"
 - o : The author's name appears to be missing or corrupted.
 - o Correction: Ensure correct citation formatting.
- "cellulosomes" (Typo)

Reviewer #2 (Comments to the Authors (Required)):

This is an interesting manuscript that describes the use of super resolution microscopy to study surface cellulosomes in the highly cellulolytic microbe *Clostridium thermocellum*. The results reveal how the surface locations of the cellulosomes change at different growth stages, and in the presence of insoluble biomass (Avicel). This is achieved by tracking the location of the CBM3A domain within CipA (the major scaffoldin). Interesting, imaging shows that the cellulosomes migrate from the cells to the biomass, effectively coating it with enzyme machinery. The positioning of unoccupied cohesin-1 (Coh1) modules are also tracked, which also change their positioning when biomass is present.

Overall, this is impressive work. My only significant criticism is that the text gets bogged down in detailed descriptions of the data analysis, so at times it is difficult to follow. This problem could be addressed by adding a paragraph to the discussion section that interprets/summarizes the data. It could clearly state the estimated number of CipA proteins present in each cellulosome complex and the estimated molecular weight of the complex; the estimated number of cellulosomes that reside on the surface when the microbe is grown in cellobiose (either at log or stationary phase); the dimensions of the cellulosome complexes and the level of variation that is observed; a description of what the data suggests about the structure/composition of the EMS interface.

Again, this is beautiful work. Below are a few questions and revisions that might improve the manuscript.

Comments/Revisions/Questions

- (1) Page 2. "Both TEM and SEM can technically achieve the resolution needed to study the cellulosome, with resolutions around 0.2 nm [18, 19] and 2 nm [20, 21] respectively." Please clarify the specific type of TEM method you're referring to. I thought cryoET methods do not reach resolutions of 0.2nm.
- (2) Page 4. "in the stationary phase . These" typo with extra space
- (3) It would good to comment on potential weaknesses/limitations of the imaging studies reported here. For example, can't fluorescence changes result from alterations in the surface accessibility of the CBM3A or Coh1 modules, and not their abundance? For CBM3A have control experiments been performed that show that the antibodies you are using are specific to CipA?
- (4) The text describing DBSCAN is a bit confusing for the non-expert. It would be good to discuss MNM and the Epsilon parameters in greater detail. They are determined through unsupervised learning? Is the dynamic range of the fluorescence

intensities sufficient to directly report on MNM? Does the epsilon radius (Eps) have a physical meaning?

(5) Figure 6. What are the units in panels C-F

(6) Figure 1. I don't think ScaE / Cthe_0736 is mentioned in the text. It would be good to add for completeness.

(7) Page 8. Figure 6. The term "MNP" is not defined. I assume it is synonymous with MNM?

(8) Page 9. "The clustering algorithm ran through an array of MNM per epsilon from 10 to 1200 at a fixed Epsilon radius set at 75 nm." What is the reasoning for selecting this as the fixed Epsilon radius value?

(9) Page 11 "The PA-GFP shows a very similar exponential or geometric distribution to that in Figure 5G" There is no figure "5G" do you mean 7G? Same issue for figure 5B cited in the text.

(10) Page 25 (Figure S4A). Image appears to show green clusters (unoccupied Cohl sites, pA-GFP-Doc) that are not substrate-attached nor co-localized with the CBM3a (red) clusters. Would be good to explain why.

Reviewer #3 (Comments to the Authors (Required)):

The manuscript is well-planned, well-executed, and well-written. It is relevant to the field as it proposes and validates a methodology to study the distribution of cellulosomes in *C. thermocellum* at different growth stages and on various substrates. Understanding the dynamics of cellulosome populations at the enzyme-microbe-substrate interface is crucial for elucidating the mechanism of cellulose deconstruction by *C. thermocellum*.

Figure 3A and 3B represent the cells in the log phase. However, there is a difference in the amount of CBM3a. Even cells in the log phase may exhibit variations in the amount of cellulosome associated with the cell.

Please clarify how it is possible to distinguish the dockerin of CipA from OlpA and OlpC.

In this example, the number of cellulosome clusters found on the cell in stationary phase with a MNM per epsilon of 200 was four clusters (6E). I suggest verifying in which figure this result was showed.

Altogether, we analyzed the normalized numbers of clusters collected from over 60 single bacterial cells in both the log and stationary phases of growth. Note that the x-axis represents the MNM per epsilon but does not give the relative size of the cluster, and the y-axis represents the normalized number of clusters per bacterial cell. The reasoning for normalizing the number of clusters per bacterial cell is to gain quantitative insight into how the density of molecules within the cellulosomal protuberances is affected during growth. During the log phase, there is an increase in the density of molecules compared to that of bacterial cells in the stationary phase. In the stationary phase, the number of identified clusters decreases as the density of molecules increases starting from 100 molecules per epsilon, whereas the maximum number of clusters per bacterial cell in the log phase is associated with 300 molecules per epsilon. Please indicate the figure showing this result.

Figure 5G shows the distribution of the normalized number of clusters versus the MNM/cluster. Is this the figure that shows this result?

The analysis of substrate-free bacteria revealed a remarkable distribution of CipA scaffoldins. The cellulosomes appear to cover the surface of the bacterial cell. Using our clustering algorithm, we can identify 14 clusters for the CipA-CBM3a containing at least 200 fluorescent molecules per Epsilon (Figure 7B). Is this the figure that shows this result?

Reviewer #1 (Comments to the Authors (Required)):

Yarborough et al provide a manuscript on super resolution fluorescence microscopy imaging of cellulosomes on the surface of *Clostridium thermocellum* cells. They show that the degree of surface coverage of cellulosomes changes under different growth conditions, and dependent on substrate contact. It is also suggested that dockerin binding sites on CipA are mostly occupied while they are not on other modules.

Overall, the manuscript is quite diffuse, as is the title. In the abstract, it is stated "providing a clearer picture of the dynamics of cellulosome populations at the enzyme microbe substrate interface." In what way do the authors think the view is now clearer? In our view, the ms would gain much if the message was sharpened, as to what new insights were obtained.

Main comments

1) I am unsure about the composition of cellulosomes, and the description in the text. Fig. 1: what is the relation between the CipA molecule on the left and the structures embraced by the bracket: does the cell have different types of CipAs bound to different "Olp" proteins within the membrane (or is it the cell wall?)? A more precise cartoon would be very helpful.

Figure 1 shows the complex nature of *C. therm* for which a precise cartoon is difficult to generate. What the figure demonstrates is how a complex CipA can attach to the surface of the bacterium (attaching probably to peptidoglycans, but still an area of debate) via the OlpB, Orf2P and SdbA binding sites (thanks to a surface layer homology domain) which the bracket is supposed to demonstrate. We also wanted to make sure that the readers realize that the CipA scaffold does not bind to the OlpA and OlpC sites. With that, we have decided to further strengthen the description of the figure with the following in the text (lines 38-49) "These scaffoldins can themselves be assembled onto secondary scaffoldins bearing type II cohesin domains, specifically through interactions with three critical anchor proteins: the heptavalent OlpB, Orf2P, and SdbA. Each of these anchor proteins contains a cohesin domain that binds with high specificity to the type II dockerin present on the primary CipA scaffoldin. OlpB, Orf2P, and SdbA not only facilitate the hierarchical assembly of a supramolecular complex that can incorporate up to 63 glycoside hydrolase enzymes but also anchor this complex firmly to the microbial cell surface via their surface layer homology (SLH) domains. The SLH domains provide attachment points within the bacterial cell wall enhancing proximity to the microbial cell. This structural organization and surface localization confer substantial advantages to cellulosome-producing microbes such as *Clostridium thermocellum*, enabling them to degrade lignocellulosic biomass more efficiently compared to organisms that rely on free cellulases."

2) Why is the green signal from PA-GFP type-I dockerin fusion protein mostly separate/distinct from the antibody-detected red fluorescence for CipA? CipA contains 9 dockerin domains, so why aren't these detected?

This is most likely because the cohesin domains are occupied by the hydrolase enzymes produced by *C. thermocellum* therefore the cohesin is occupied and the doc-GFP protein cannot bind. There is a probability that not all cohesin domains are occupied. In this instance the two fluorescence signals are distinct.

The authors state "Surprisingly, it seems that most empty type-I cohesins are located on the bacterial cell wall and that the CipA scaffoldins are almost fully saturated with enzymes." That seems very unlikely, except if the Olp dockerin domains have an enormously higher dissociation constant (i.e. turnover) than

those on the CipA protein. Is there evidence for this? If not, the authors should do a pulse chase, in which they add the PA-GFP dockerin fusion and wash it off, and do imaging without fixation (please see below - not clear how any imaging experiment was done).

Thank you for this comment. We actually do not know that much about the dissociation constant for the Olp cohesin. It could well be that these interactions are weaker. For now, this is just an observation, so we have changed “Surprisingly, it seems that most empty type-I cohesins are located on the bacterial cell wall and that the CipA scaffoldins are almost fully saturated with enzymes” by “Surprisingly, most empty type-I cohesins are located on the bacterial cell wall” (lines 167-168).

3) Fig. 2A - it seems obvious where the outlines of the cells are, but not completely clear. Please draw outlines with an e.g. thin white line, or make a cartoon underneath the FM image, if there is no appropriate bright field image available for the image.

We appreciate the reviewers' concerns. However, we do not have the white light image for this image and we feel that making a cartoon would be a little arbitrary. We believe the figure serves its purpose and may be left as is. We hope the reviewer will agree.

4) The manuscript does not even contain page numbers, let alone line numbers. That's quite a pain for any reviewer.

We apologize for the oversight. We have added page numbers as well as the number lines.

Section II, 1. "The main drawback of traditional optical microscopy is the inability to systematically quantify the information within the image. Unfortunately, with optical images, it is difficult to quantify the fluorescence from each bacterium, let alone analyze any significant portion of the bacterial cells." Why would that be? There are plenty of programs out there to automatically quantify fluorescence of fields of cells, or within single cells ("Mircobetracker", for example). Please give the educated reader appropriate reasoning why you believe in what is stated.

The reviewer raises a valid point that numerous software tools exist for automated quantification of fluorescence in optical microscopy images, both at the field and single-cell levels. Indeed, these tools allow researchers to measure integrated or mean fluorescence intensities within defined regions of interest corresponding to bacterial cells. However, the fundamental limitation of traditional far-field optical microscopy, particularly when aiming for precise and systematic quantification of the true amount of fluorescence originating from individual fluorophores within each bacterium, stems from the diffraction limit of light. This limitation introduces several key challenges that can obscure or distort the actual fluorescence signal:

1. **Ensemble Averaging within the Diffraction Limit:** Traditional optical microscopy cannot resolve features smaller than approximately half the wavelength of light used. In the context of a bacterium labeled with multiple fluorescent molecules, the signal observed from a single diffraction-limited spot is an *ensemble average* of the fluorescence emitted by all the fluorophores within that volume. This means:
 - **Loss of Individuality:** We cannot distinguish and count individual fluorophores. A brighter spot could represent more fluorophores, but it could also represent a few brighter fluorophores or a cluster of moderate brightness.
 - **Inability to Account for Heterogeneity:** Even if we measure the integrated intensity of a diffraction-limited spot associated with a bacterium, we only get an average signal. We

lose information about the distribution and stoichiometry of the fluorescent labels within that cell. Some bacteria might have more labeled targets than others, or the spatial organization of these targets might vary significantly, but the diffraction limit blurs these details.

- **Potential for Misinterpretation:** Changes in the observed fluorescence intensity could be due to changes in the number of labeled molecules, their local environment affecting brightness, or even just their spatial reorganization within the diffraction-limited volume.
2. **Challenges in Accurate Background Subtraction and Autofluorescence Correction:** While software algorithms can perform background subtraction, the presence of out-of-focus light and autofluorescence within the diffraction-limited volume surrounding a bacterium can contribute significantly to the measured signal. It becomes challenging to precisely disentangle the true signal from the specific labels from these confounding factors at the single-molecule level.
 3. **Limited Resolution for Structural Context:** Traditional optical microscopy often lacks the resolution to define the boundaries of small bacterial cells precisely or to resolve internal structures where the fluorescent labels might be localized. This can lead to inaccuracies in defining the region of interest for quantification and potential inclusion of signal from outside the cell or exclusion of signal from within.
 4. **Photobleaching Effects:** While photobleaching affects all fluorescence microscopy techniques, in traditional widefield or confocal microscopy, the simultaneous excitation of multiple fluorophores within a diffraction-limited volume makes it harder to track the bleaching of individual molecules and correct for its impact on the overall intensity measurement.

Super resolution techniques like STORM and PALM overcome the diffraction limit by employing strategies to sequentially and sparsely activate and image individual fluorophores. This allows for:

1. **Single-Molecule Localization and Counting:** By ensuring that only a small subset of fluorophores emit at any given time, their individual point spread functions (PSFs) can be resolved and their positions determined with nanometer precision. This directly enables the counting of individual fluorescent molecules.
2. **True Quantification of Fluorophore Number:** Instead of measuring an ensemble average, STORM and PALM allows for the direct estimation of the number of fluorophores within a bacterium by counting the number of localized events over time, taking into account blinking and photobleaching kinetics.
3. **Mapping Molecular Distributions with High Precision:** STORM/PALM's high spatial resolution allows for high-precision mapping of the location and distribution of the fluorescently labeled molecules within the bacterial cell. This high precision provides crucial information about their organization and stoichiometry in relation to cellular structures.
4. **Reduced Impact of Out-of-Focus Light:** The sequential imaging and localization approach in STORM/PALM inherently reduces the contribution of out-of-focus light to the final reconstructed image and the quantification of individual fluorophores.

Therefore, while traditional optical microscopy with automated analysis tools can provide valuable information about the average fluorescence intensity of bacterial populations or individual cells, it fundamentally struggles to provide a *true* and systematic quantification of the number and distribution of individual fluorescent molecules within those cells due to the diffraction limit. STORM and PALM offer a technique that allows us to achieve this level of quantitative precision and provides better insights into the molecular architecture and function of bacteria that are often inaccessible with conventional techniques

Therefore we are adding the following paragraph to the main text to help emphasize this point:

“Traditional far-field optical microscopy, even with automated analysis tools, is limited by the diffraction limit of light, which prevents precise quantification of individual fluorophores within bacterial cells. This limitation causes averaging of fluorescence signals, loss of molecular detail, difficulty accounting for heterogeneity, and susceptibility to background, autofluorescence, and photobleaching effects, therefore, restricting our ability to quantify the fluorescence from each bacterium, let alone analyze any significant portion of the bacterial cells. Super-resolution methods like STORM and PALM overcome these challenges by localizing and counting individual fluorophores with nanometer precision, enabling accurate quantification, high-resolution mapping of molecular distributions, and reduced out-of-focus light interference. Because a portion of the data generated from the super resolution images contains the x-y coordinates of the calculated gaussian for each fluorescent event, we can use this information to quantify the location and number of fluorescent events within a region of the image using unsupervised machine learning. This type of machine learning, also referred to as cluster analysis, allows us to define a certain epsilon radius (Eps) and set a minimum number of points, or in this case, the number of fluorescent molecules within the defined radius. The clustering algorithm chosen for this work was the density-based spatial clustering of applications with noise (DBSCAN), which is a propagative cluster detection method linking points that are closely packed together and detecting outliers using the user-defined epsilon radius and a minimum number of points or molecules [30]. Some of the main benefits of using DBSCAN for single molecule experiments are that DBSCAN can detect arbitrary shaped clusters, is quick, and is robust to outliers [30]. For the work presented here, we initially decided to set the epsilon radius to 75 nm for the CBM3a-tagged Alexa Fluor 647 molecules.”

5) As I take it, a main objective of the manuscript is to provide a protocol for super resolution imaging of cellulosomes. However, there is not a single line in the methods about imaging. Please fill in how STORM and PALM was done, what were the settings.

The objective of this manuscript was to perform, for the first time, a quantitative analysis of cellulosome distribution on both the bacterial cell surface and avicel substrates. Given that STORM and PALM imaging techniques have been extensively discussed in prior literature, our study instead emphasizes elucidating the potential mechanisms employed by *Clostridium thermocellum* in biomass degradation. Nevertheless, we have added some text in the materials in methods about the way the experiments were conducted as well as external references

Experimental method

C. thermocellum cultures were grown in serum bottles containing MTC medium supplemented with either cellobiose or Avicel with each serum bottle dedicated to a single biological replicate. Samples were harvested during both logarithmic and stationary growth phases.

For imaging preparation, 0.5 mL aliquots were withdrawn and washed in 50 mM sodium acetate buffer containing 2 mM CaCl₂ (sample preparation buffer) to remove residual medium components and free cellulosomes. These washed cells were incubated for 10 min with 0.75 µg/µL CBM3a antibody, 0.387 µg/µL Alexa Fluor 647, and 0.387 µg/µL photoactivatable GFP (PA-GFP). Following labeling, samples were washed three times with the same sample preparation buffer. Labeled cells were then immobilized onto [specify coverslip type, e.g., No. 1.5H high-precision] coverslips.

For STORM imaging, the imaging buffer consisted of phosphate-buffered saline (PBS) supplemented with 0.5 mg/mL glucose oxidase, 100 mM β-mercaptoethylamine (MEA), and ~10 µL potassium chloride, prepared fresh prior to imaging to ensure optimal photoswitching conditions.

Imaging

Super-resolution imaging was performed on a Zeiss [model] microscope equipped with 405 nm, 488 nm, and 641 nm laser lines. Alexa Fluor 647-labeled cellulosomes were initially imaged using 641 nm excitation, followed by sequential imaging of PA-GFP-labeled targets using 405 nm photoactivation and 488 nm excitation. Image acquisition was performed at 30 ms exposure time per frame, with 15,000 consecutive frames collected per channel to ensure adequate sampling for high-fidelity STORM/PALM reconstruction.

6) Text above Fig. 2 "with minimal sample preparation and disruption of the bacterium, with a resolution of 35 nm." Please show how you arrived at this resolution, and how sample preparation was done (see point 5). Please provide evidence that the red signals for the STORM signals are composed of single molecules (single blinking events?). The images don't look like super resolution (SR) images to me, except for the PA-GFP signals. Speaking of SR - I have so far not hear "super high resolution", just either super - or high resolution. I seems very peculiar that the authors would try to top the term "SR" imaging.

We have replaced all instances of super high resolution by super resolution

We approached the resolution question for the super resolution imaging using the common method for estimating the resolution by splitting the detected localization between two images and running a Fourier Ring Correlation between the two images. Fourier Ring Correlation (FRC) is an objective, data-driven method to estimate resolution in super-resolution microscopy. The dataset is split into two independent halves, each reconstructed separately. Their Fourier transforms are compared ring-by-ring across spatial frequencies. Correlation is high at low frequencies and falls off at higher ones as noise dominates. The resolution is defined at the frequency where correlation drops below a standard threshold often the 1/7 criterion.

FRC directly reflects the achievable resolution of the actual experiment, accounting for photon statistics, labeling density, and noise, rather than relying on beads or localization precision alone. This ensures that reported resolution values are both robust and representative of the biological dataset analyzed.

The STORM signal does not resemble a typical super-resolution image because the cellulosomes are extremely densely packed. Even with the enhanced resolution provided by STORM, the spatial separation between individual cellulosomes remains below the optical resolvability limit, preventing them from appearing as distinct structures.

Below, please find mostly textual comments.

Result 1

1. "cell-free cellulosomes" could be "free cellulosomes"
o "Cell-free cellulosomes" is somewhat redundant because "cellulosomes" are defined as extracellular structures, and sounds like cellulosomes arose independent from cells. "Free cellulosomes" would make more sense.

Thank you for this suggestion, we have made the necessary changes for better clarification

2. "It was further proposed that detachment of the bacterial cells may be connected to a controlled release of the cells from the cellulose-bound cellulosome, which would continue to hydrolyze the

substrate."

o : unclear phrasing

Thank you for pointing this out, we have rewritten the sentence for better clarity with the following new sentence "These studies also noted a reduction of cell-associated cellulosomes in the stationary phase of cell growth compared with the log phase. It was further proposed that detachment of the bacterial cells may be connected to a controlled release of the cells from the cellulose-bound cellulosome, allowing the continuation of hydrolysis of the substrate without the need for the bacteria cell to be present on the substrate." (Lines 122-127)

3. *"examined at nanoscale resolution single C. thermocellum cells"*

o : Incorrect word order; should be "examined single C. thermocellum cells at nanoscale resolution."

Thank you for this suggestion, we have made the correction

4. *"observed phenotypic cellulosomal heterogeneity mediated by soluble sugar concentration in the media"*

o : Incorrect use of "mediated by." The soluble sugar concentration is likely influencing or regulating, not "mediating."

Thank you for this suggestion, we have made the correction

5. *"suggesting a division-of-labor strategy"*

o : "Division of labor" does not require hyphens unless used as a compound adjective.

Thank you for this suggestion, we have made the correction

6. *"further clarify the status of cellulosomes with regards to the presence of both the primary scaffoldin and the cellulosomal enzymes on cells of C. thermocellum and on the substrate"*

o : "With regards to" could be "with regard to"

Thank you for this suggestion, we have made the correction

7. *"we used a primary antibody targeting the CBM3a of the CipA scaffoldin with a secondary antibody fused to Alexa Fluor 647"*

o Unclear sentence. Use "and" instead of "with" or any other modification to clarify what was done

Thank you for this suggestion, we have made the correction

8. *"a PA-GPF" Should be "a PA-GFP"*

o : Typographical error ("PA-GPF" instead of "PA-GFP").

Thank you for this suggestion, we have made the correction

9. *"grown on cellobiose in the stationary phase ."*

o : Extra space before the period.

Thank you for this suggestion, we have made the correction

10. *"we were able to further validate these original findings, with minimal sample preparation and disruption of the bacterium, with a resolution of 35 nm."*

o : Too many instances of "with," making the sentence clunky.

Thank you for the suggestion, we have modified the sentence to *"Using STORM and PALM imaging, we were able to further validate these original findings using minimal sample preparation and disruption of the bacterium with a resolution of 35 nm."*

11. *"To date, these cellulosomal protuberances and their dynamics have not been fully characterized, especially on process-relevant feedstocks."*

o : "Process-relevant feedstocks" is unclear. It would be better to specify what type of feedstocks (e.g., lignocellulosic biomass).

Thank you for this suggestion, we have edited the sentence as follows *"To date, these cellulosomal protuberances and their dynamics have not been fully characterized, especially on process-relevant lignocellulosic biomass"*

12. *"These advanced optical microscopy techniques appear to be particularly well suited to the quantitative study of the population of cellulosomes and their dynamics over time during biomass deconstruction."*

o : "Study of the population of cellulosomes", Better phrased as "quantitative analysis of cellulosome populations and their dynamics during biomass deconstruction."

We have made this change

1. *"cell-free cellulosomes on the surface of cellulose"*

o : Cellulosomes are usually attached to cells or substrates. If they are truly cell-free, clarification is needed.

We have made the corrections as suggested and removed the cell from cell-free cellulosomes to free cellulosome

2. *"Bayer and coworkers discovered these protuberances on C. thermocellum and reported that both their size and longitudinal arrangement varied as determined by either a selective cationic electron-dense probe or antibodies specific to the CipA scaffoldin"*

o : The phrase "as determined by" is misleading because the size and arrangement of protuberances do not depend on the labeling technique-they were observed using these methods.

We have changed the language to the following *"Bayer and coworkers discovered these protuberances on C. thermocellum and reported that both their size and longitudinal arrangement varied as demonstrated by either a selective cationic electron-dense probe or antibodies specific to the CipA scaffoldin"*

Result 2

• "From these data, there is a distinct change in the pattern of the polycellulosomal protuberances which form highly decorated and interconnected protuberances on the bacterial cells in the log phase."

o : Redundant phrasing ("polycellulosomal protuberances" vs. "highly decorated and interconnected protuberances"). Also, "From these data, there is" is an awkward construction.

Thank you for this suggestion, we have made the necessary changes

- *"These new structures appear to be depleted when the microbe is in the stationary phase of growth."*
o : "Appear to be" is unnecessary unless there is doubt. "Depleted" is better suited for resources rather than physical structures.

Thank you for this suggestion, we have modified the sentence as follows "These new structures are absent when the microbe is in the stationary phase of growth"

- *"Surprisingly, it seems that most empty type-I cohesins are located on the bacterial cell wall and that the CipA scaffoldins are almost fully saturated with enzymes."*
o : "It seems that" weakens the statement.

Thank you for this suggestion, we have modified the sentence.

- *"We utilized the DBSCAN cluster analysis with the number of clusters per bacterial cell versus the MNM per epsilon."*
o : The sentence is incomplete; it does not clarify what was being analyzed.
o Wordy Phrasing

Thank you for this suggestion, we have modified the sentence as follows "To get a better understanding of the distribution of the CipA scaffolds, DBSCAN cluster analysis was used to analyze the images using a combination of the number of clusters per bacterial cell versus the MNM per epsilon."

- *"Red fluorescence denotes CBM3a (CipA scaffoldin)-associated label, whereas green fluorescence indicates unoccupied type I cohesin-related sites (see legend to Figure 2 and text for more detail)."*
o : "See legend to Figure 2" → should be "See the legend in Figure 2."

We have made the correction

- *"We also give a visual example of the number of clusters and their overall size on a specific bacterial cell in the log phase and in the stationary phases of growth."*
o : "We also give" is informal. "Specific bacterial cell" is redundant.

We have made the following correction to the sentence "A visual example is given with the number of clusters and their overall size on a bacterial cell in the log phase and in the stationary phases of growth. In this example, the number of cellulosome clusters found on the cell in stationary phase with a MNM per epsilon of 200 was four clusters (6E), and the number of clusters found for the bacterial cell in the log phase for 200 MNM per epsilon was nine (6C). "

- *"For example, for 500 MNM per epsilon, two nodules were detected in the stationary phase, but eight*

nodules were detected in the log phase, thus illustrating that the nodules on the bacterial cells are larger in the log phase."

o : Repetitive use of "nodules" and "log phase."

We have replaced the sentence with the following "For example, for 500 MNM per epsilon, two nodules were detected in the stationary phase, but eight were detected in the log phase, thus illustrating that the nodules on the bacterial cells are larger in the log phase. "

- *"Altogether, we analyzed the normalized numbers of clusters collected from over 60 single bacterial cells in both the log and stationary phases of growth."*

o : "Single bacterial cells" is redundant (all bacterial cells are single unless otherwise stated).

We have edited the sentence to the following "Altogether, we analyzed the normalized numbers of clusters collected from over 60 bacterial cells in both the log and stationary phases of growth."

- *"The reasoning for normalizing the number of clusters per bacterial cell is to gain quantitative insight into how the density of molecules within the cellulosomal protuberances is affected during growth."*

o : "The reasoning for" is clunky.

We have replaced the sentence with "Normalizing the number of clusters per bacterial cell gains quantitative insight into how the density of molecules within the cellulosomal protuberances is affected during growth. "

- *"The signal for the unoccupied type-I cohesins and these analyses indicate that they are fairly dispersed on the surface and not co-localized with cellulosomes, which indicates either a low abundance of unoccupied type-I cohesins or a low abundance of type-I cohesins on the surface of these bacteria."*

o : Confusing conclusion-does it mean unoccupied cohesins are low in number, or that they are not clustering?

We have modified the sentence as follows "The signal for the unoccupied type-I cohesins and these analyses indicate that they are fairly dispersed on the indicating either a low abundance of unoccupied type-I cohesins or a low abundance of type-I cohesins on the surface of these bacteria."

In result 3: Rephrase this sentence, the idea is vaguely expressed

"When analyzing the stationary phase when attached to Avicel, we found a significant amount of colocalization of the CipA-CBM3a and the type-I cohesin, which is primarily located on the Avicel particle and in particular at the point where the bacterium is attached to the Avicel particle (Figure 7H-J)."

The paragraph is mostly well-structured, but there are a few grammatical and stylistic issues. Here are the key mistakes:

1. *"When analyzing the stationary phase when attached to Avicel"*

o Issue: Awkward phrasing and unclear subject. "When analyzing" implies that "we" (the researchers) are the subject, but "the stationary phase when attached to Avicel" is unclear. The stationary phase itself is not "attached"-the bacteria are.

2. *"found a significant amount of colocalization"*

o Issue: "Amount" is generally used for uncountable nouns, but "colocalization" is more abstract. "Significant degree of colocalization" is a better phrasing.

3. *"the CipA-CBM3a and the type-I cohesin"*

o Issue: The article "the" before "CipA-CBM3a" is unnecessary

4. *"which is primarily located on the Avicel particle"*

o Issue: The relative pronoun "which" refers to "type-I cohesin," but the sentence makes it sound like both "CipA-CBM3a and type-I cohesin" share this location. This is ambiguous.

5. *"and in particular at the point where the bacterium is attached to the Avicel particle"*

o Issue: "And in particular" is redundant. Also, "the bacterium" implies only one, but it seems to refer to bacteria in general.

To address all of these concerns within Result 3, we have adjusted this sentence as follows "When analyzing the bacteria that are attached to Avicel in stationary phase, there is a significant degree of colocalization of CipA-CBM3a and the type-I cohesin, primarily located on the Avicel particle at the point where the bacteria is attached to the Avicel particle (Figure 7H-J)."

Result 4

• *"The cellulosomes appear to cover the surface of the bacterial cell."*

o : "Appear to" is unnecessary unless there is uncertainty.

We have adjusted the sentence to "The analysis of substrate-free bacteria revealed a remarkable distribution of CipA scaffoldins. The cellulosomes cover the surface of the bacterial cell. Using our clustering algorithm, we can identify 14 clusters for the CipA-CBM3a containing at least 200 fluorescent molecules per Epsilon (Figure 7B) but only eight clusters for the PA-GFP containing at least 10 fluorescent molecules per Epsilon (Figure 7C)."

• *"The PA-GFP shows a very similar exponential or geometric distribution to that in Figure 5G, demonstrating that the combination of unoccupied OlpA/OlpC/CipA sites is similar for detached and attached bacteria."*

o : "A very similar exponential or geometric distribution" is unclear. Is it both, or one?

Yes, this needed to be clarified, we have removed the geometric distribution because the distribution of the OlpA/OlpC/CipA is actually more exponential.

• *"However, it appears that the distribution of the CipA-CBM3a clusters for the detached bacteria is quite different from the ensemble distribution shown in Figure 5G, which would indicate more of a mixed population in the culture."*

o : "It appears that" weakens the statement. "Would indicate" is also weak.

We have corrected the sentence with the following "However, the distribution of the CipA-CBM3a clusters for the detached bacteria is quite different from the ensemble distribution shown in Figure 8G, indicating more of a mixed population in the culture."

• *"The overall intensity and distribution pattern are significantly different and show a loss in the number of cellulosomes attached to the surface of stationary-phase bacteria (Figure 7E)."*

o : "Loss in the number of cellulosomes" is awkward.

Regarding the previous comments, we have changed the sentence as follows “The PA-GFP shows a very similar exponential or geometric distribution to that in Figure 5G, demonstrating that the combination of unoccupied OlpA/OlpC/CipA sites is similar for detached and attached bacteria. However, the distribution of the CipA-CBM3a clusters for the detached bacteria is quite different from the ensemble distribution shown in Figure 5G, indicating more of a mixed population in the culture. In the stationary phase, our analysis of detached bacterial cells revealed a different pattern than that observed for log-phase cells”

2. Wordy Phrasing

- *"Figures 7F and G show representative distributions for an MNM of 500 for CipA-CBM3a and 50 for the PA-GFP with 17 clusters detected for CipA-CBM3a and only seven clusters detected for PA-GFP (Figure 7H)."*

o : Too much information in one sentence.

We now have two sentences which are “Figures 7F shows the representative distribution of for the MNM of 20 for CipA-CBM3a. Figure 7G shows representative distributions for an MNM of 10 for the PA-GFP with 17 clusters detected for CipA-CBM3a and only seven clusters detected for PA-GFP (Figure 7H).”

Discussion

- *"on either soluble and insoluble substrates"*

o : Incorrect use of "either" with "and." "Either" should be followed by "or," not "and."

We corrected the sentence with “Even though it appears that cells shed cellulosomes during growth on either soluble or insoluble substrates, it is clear from Figures 4, 7 and 9 that these cellulosomes are targeted to the biomass at the contact points in a concerted manner when substrate is present”

- *"attached or attached to biomass"*

o : Repetition-either "attached" was mistakenly repeated, or "detached" was intended.

We corrected the sentence with “It is important to note that in the present study these conclusions were drawn from average fluorescence values, corresponding to cellulosomes from tens of bacteria grown on different substrates, stages of growth, and either is or is not attached to biomass.”

- *"when the microbe was actively engaged in degradation of cellulosic biomass"*

o : "Engaged in degradation of" is unnatural. The verb "degrading" is more concise.

We have replaced the following new sentence “However, the distribution of these protuberances was different in cells grown to the log and stationary phases, when the microbe was degrading the cellulosic biomass.”

- *"Additionally, there seems to be increased colocalization of empty type-I cohesin and cellulosomes at these contact points"*

o : "There seems to be" is weak and redundant.

We have reworked the sentence to be: “Additionally, there is an increased colocalization of empty type-I cohesin and cellulosomes at these contact points (Figure 4), indicating that some of the

dockerin-bearing enzyme molecules occupying these cohesin-bearing scaffoldins were released to interact directly with the substrate.”

- *"The unoccupied type-I cohesin can be found on different protein scaffoldins, e.g., OlpA and OlpC, attached to the microbial cell wall, but also on the primary scaffoldin, CipA, either detached or initially bound to the bacterial cell wall via the secondary scaffoldin OlpB (Figure 1)."*

o : The phrase "but also" disrupts parallelism. The sentence is too long and convoluted.

We have worked on improving this sentence with the following two sentences “The unoccupied type-I cohesin can be found on different protein scaffoldins, e.g., OlpA and OlpC, attached to the microbial cell wall. These structures are also found on the primary scaffoldin, CipA, and can either detached or initially bound to the bacterial cell wall via the secondary scaffoldin OlpB (Figure 1)”

2. Wordy Phrasing

- *"This process is even more challenging to understand when considering cellulosomal microbes that primarily use an intricate multienzyme complex to deconstruct cellulosic substrates instead of free enzymes."*

o : Wordy and slightly redundant.

We have adjusted the sentence as follows “Biomass deconstruction by thermophilic microbes is a complex and heterogeneous process. This process is even more challenging to understand due to the intricate multienzyme complex used by these cellulosomal microbes to deconstruct cellulosic substrates.

“

- *"We made the same observation when C. thermocellum was grown on both soluble and insoluble substrates, which seems to indicate that the loss of cellulosome is not triggered by their attachment to insoluble substrates."*

o s:

1. "Loss of cellulosome" should be plural ("cellulosomes").
2. "Their" does not agree in number with "cellulosome."

We have made the correction

- *"It is also expected that unoccupied type-I cohesins on CipA would be found close to the CipA-CBM3a and are therefore considered to be colocalized."*

o : "Expected" and "considered" introduce unnecessary uncertainty.

We have adjusted the sentence as follows “It is also expected that unoccupied type-I cohesins on CipA would be found close to the CipA-CBM3a and colocalized.”

3. Redundancy & Repetition

- *"Indeed, log-phase bound and detached cells appear to exhibit the same distribution profiles for the number of molecules per cellulosome cluster."*

o : "Appear to exhibit" is wordy.

We have removed the words “Appear to”

- *"One notable phenomenon that occurs in the stationary phase of growth is the sheer saturation of*

substrate with cellulosomes."

o : "Sheer saturation" is redundant-saturation already implies completeness.

We have removed the word Sheer from the sentence

- "Again, this phenomenon is consistent with early TEM studies [3, 15] and could potentially explain some of the slowdowns that happen during growth of *C. thermocellum* on insoluble substrates, whereby the saturation of the substrates with cellulosomes could be counterproductive and lead to 'traffic jams' of the cellulases on the substrate."

o s:

1. "Again" is unnecessary.

Removed Again

2. "Could potentially" is redundant-use either "could" or "potentially."

Removed potentially

3. "Slowdowns that happen during growth" can be simplified.

Simplified this sentence

4. Ambiguity & Logical Issues

- "These results are consistent with early TEM observations [3, 15] whereby the exocellular protuberances of cellulose-bound cells were reported to undergo dramatic conformational change to form contact corridors where the cellulosomes are concentrated on the surface of the cellulosic substrate." o : "Whereby" is improperly used-use "in which."

We have replaced Whereby with "in which"

- "This study highlights the potential of using these new optical techniques, combined with specific mutations in *C. thermocellum* or other cellulosomal microbes, to probe and challenge other hypotheses that have been proposed regarding cellulosomes and their mode of action, such as the role of 'shuttle' scaffoldins and the role of the attachment of *C. thermocellum* to biomass."

o : "The role of the attachment of *C. thermocellum* to biomass" is awkward.

We have removed "The Role of"

5. Formatting Issues & Typos

- "[tal] et al [29]"

Fixed

Reviewer #2 (Comments to the Authors (Required)):

This is an interesting manuscript that describes the use of super resolution microscopy to study surface

cellulosomes in the highly cellulolytic microbe Clostridium thermocellum. The results reveal how the surface locations of the cellulosomes change at different growth stages, and in the presence of insoluble biomass (Avicel). This is achieved by tracking the location of the CBM3A domain within CipA (the major scaffoldin). Interesting, imaging shows that the cellulosomes migrate from the cells to the biomass, effectively coating it with enzyme machinery. The positioning of unoccupied cohesin-1 (Coh1) modules are also tracked, which also change their positioning when biomass is present.

Overall, this is impressive work. My only significant criticism is that the text gets bogged down in detailed descriptions of the data analysis, so at times it is difficult to follow. This problem could be addressed by adding a paragraph to the discussion section that interprets/summarizes the data. It could clearly state the estimated number of CipA proteins present in each cellulosome complex and the estimated molecular weight of the complex; the estimated number of cellulosomes that reside on the surface when the microbe is grown in cellobiose (either at log or stationary phase); the dimensions of the cellulosome complexes and the level of variation that is observed; a description of what the data suggests about the structure/composition of the EMS interface.

Thank you for the kind words. To help clarify the results from the complex data analysis, we added the following paragraph in the discussion section "This super-resolution imaging and DBSCAN clustering analysis revealed that each cellulosome complex contains densely packed CipA scaffoldins, corresponding to tightly organized macromolecular assemblies. When grown on cellobiose, *C. thermocellum* cells in the log phase displayed more numerous and larger cellulosome clusters on their surfaces compared to stationary phase, with CipA scaffoldins largely saturated with enzymes and few unoccupied cohesins. On insoluble Avicel, log-phase cells retained cellulosomes primarily on the cell surface, whereas stationary-phase cells relocated large clusters to the EMS interface, saturating the biomass surface. Detached cells in stationary phase showed marked depletion of surface cellulosomes compared to bound cells, but unoccupied type-I cohesin distribution remained constant across growth phases and conditions. These results indicate dynamic redistribution of cellulosomes during growth, with a functional shift toward substrate-associated degradation in stationary phase, while cohesin vacancies remain largely unaffected."

Again, this is beautiful work. Below are a few questions and revisions that might improve the manuscript.

Comments/Revisions/Questions

(1) Page 2. "Both TEM and SEM can technically achieve the resolution needed to study the cellulosome, with resolutions around 0.2 nm [18, 19] and 2 nm [20, 21] respectively." Please clarify the specific type of TEM method you're referring to. I thought cryoET methods do not reach resolutions of 0.2nm.

We have added the term conventional before TEM and SEM, we are simply comparing traditional optical microscopy to the conventional TEM and SEM to compare the overall resolution. Thank you for asking for the clarification.

(2) Page 4. "in the stationary phase . These" typo with extra space

This was addressed

(3) It would good to comment on potential weaknesses/limitations of the imaging studies reported here.

For example, can't fluorescence changes result from alterations in the surface accessibility of the CBM3A or Coh1 modules, and not their abundance? For CBM3A have control experiments been performed that show that the antibodies you are using are specific to CipA?

Yes, we have performed labeling experiments that determined that the antibody used for the CBM on CipA has a high specific binding to the CBM. This involved looking at different enzymes, bacterial systems to verify the specificity.

(4) The text describing DBSCAN is a bit confusing for the non-expert. It would be good to discuss MNM and the Epsilon parameters in greater detail. They are determined through unsupervised learning. Is the dynamic range of the fluorescence intensities sufficient to directly report on MNM? Does the epsilon radius (Eps) have a physical meaning?

Thank you for this suggestion, we have added the following new paragraph to the paper giving more detail on the epsilon radius and the MNM.

Figure 5 illustrates the DBSCAN algorithm with the green spheres representing the core points considered part of a cluster as they fulfill two criteria: 1) they are located within the epsilon radius and 2) they contain at least the minimum number of molecules (MNM). The epsilon radius is defined as the maximum center-to-center distance within which two localized molecules are considered spatial neighbors. In our analysis, we set the epsilon radius to 75nm, a value chosen to reflect the estimated upper bound of distances between one CBM3a-binding sites within a single cellulosome and another CBM3A-binding sites within another single cellulose based on structural models and to account for localization precision, allowing for a direct physical interpretation in the context of cellulosome organization. For the MNM parameter, this parameter specifies the smallest number of neighboring molecules within the epsilon radius in order to classify a localization as a core molecules. The value of the MNM was selected empirically, taking into account the labeling density and localization uncertainties. Molecules that meet both the epsilon radius and the MNM criteria are designated as core molecules of the same Cluster.

(5) Figure 6. What are the units in panels C-F - need to add the scale bar to the images.

We have added the scale bar to the figures.

(6) Figure 1. I don't think ScaE / Cthe_0736 is mentioned in the text. It would be good to add for completeness.

We added this information in the introductory paragraph as follows "These scaffoldins can themselves be assembled onto secondary scaffoldins bearing type II cohesin domains, specifically through interactions with three critical anchor proteins: the heptavalent OlpB, Orf2P, and SdbA. They can also be assembled on a scaffold that is not cell associated, ScaE. Each of these anchor proteins contains a dockerin domain that binds with high specificity to the type II cohesins present on the primary CipA scaffoldin. OlpB, Orf2P, and SdbA not only facilitate the hierarchical assembly of a supramolecular complex that can incorporate up to 63 glycoside hydrolase enzymes but also anchor this complex firmly to the microbial cell surface via their surface layer homology (SLH) domains. The SLH domains provide robust attachment points within the bacterial cell wall, significantly enhancing cellulosome stability, proximity to the microbial cell, and overall efficiency. This structural organization and surface localization confer substantial advantages to cellulosome-producing microbes such as *Clostridium thermocellum*, enabling them to degrade lignocellulosic biomass more efficiently compared to organisms that rely on free cellulases."

(7) Page 8. Figure 6. The term "MNP" is not defined. I assume it is synonymous with MNM? Need to fix this image.

We have fixed the image, thank you for pointing this out

(8) Page 9. "The clustering algorithm ran through an array of MNM per epsilon from 10 to 1200 at a fixed Epsilon radius set at 75 nm." What is the reasoning for selecting this as the fixed Epsilon radius value?

We choose this value because the extended length of a CipA is ~150nm, therefore, we wanted to see how many CipAs could fit within this radius.

(9) Page 11 "The PA-GFP shows a very similar exponential or geometric distribution to that in Figure 5G" There is no figure "5G" do you mean 7G? Same issue for figure 5B cited in the text.

That description was meant for figure 8 and we have taken care of that, thank you for pointing this out and we have made the necessary changes.

(10) Page 25 (Figure S4A). Image appears to show green clusters (unoccupied Cohl sites, pA-GFP-Doc) that are not substrate-attached nor co-localized with the CBM3a (red) clusters. Would be good to explain why.

Thank you for this questions, within PALM/STORM imaging, non-specific fluorescence can arise from residual contaminants or adsorbed fluorophores on the glass substrate, producing spatially random localizations that do not correspond to biological structures. To prevent these artifacts from influencing our quantitative analysis, DBSCAN clustering was performed exclusively on regions of interest containing bacterial cells or Avicel particles, with localizations from the bare glass surface excluded.

Reviewer #3 (Comments to the Authors (Required)):

*The manuscript is well-planned, well-executed, and well-written. It is relevant to the field as it proposes and validates a methodology to study the distribution of cellulosomes in *C. thermocellum* at different growth stages and on various substrates. Understanding the dynamics of cellulosome populations at the enzyme-microbe-substrate interface is crucial for elucidating the mechanism of cellulose deconstruction by *C. thermocellum*.*

Figure 3A and 3B represent the cells in the log phase. However, there is a difference in the amount of CBM3a. Even cells in the log phase may exhibit variations in the amount of cellulosome associated with the cell.

Please clarify how it is possible to distinguish the dockerin of CipA from OlpA and OlpC.

Unfortunately, we cannot distinguish between the Cipa, OLPA and OIPC when the fluorescence is coming from the bacteria cell because they all have type 1 cohesins meaning that our doc1-GFP can bind to any of these. This is the limitation of our current suite of fluorescent probes.

In this example, the number of cellulosome clusters found on the cell in stationary phase with a MNM per epsilon of 200 was four clusters (6E). I suggest verifying in which figure this result was showed.

The four clusters were image 6D and we have corrected the typo.

Altogether, we analyzed the normalized numbers of clusters collected from over 60 single bacterial cells in both the log and stationary phases of growth. Note that the x-axis represents the MNM per epsilon but does not give the relative size of the cluster, and the y-axis represents the normalized number of clusters per bacterial cell. The reasoning for normalizing the number of clusters per bacterial cell is to gain quantitative insight into how the density of molecules within the cellulosomal protuberances is affected during growth. During the log phase, there is an increase in the density of molecules compared to that of bacterial cells in the stationary phase. In the stationary phase, the number of identified clusters decreases as the density of molecules increases starting from 100 molecules per epsilon, whereas the maximum number of clusters per bacterial cell in the log phase is associated with 300 molecules per epsilon. Please indicate the figure showing this result.

We added the following sentence to the paragraph in hopes that this explains that we ran the analysis on 60 individual bacteria cells and did not show all 60 images. The selection process was simply using python to capture the coordinates of each bacterium and running the quantification script on each of the individual bacteria. We used Figure 6 to show an example of a log phase and of a stationary phase bacterium. Here is the updated sentence “Altogether, we analyzed the normalized numbers of clusters collected from over 60 bacterial cells (not shown) in both the log and stationary phases of growth with an example of one bacteria cell in log phase (figure 6A) and stationary phase (figure 6B).”

Figure 5G shows the distribution of the normalized number of clusters versus the MNM/cluster. Is this the figure that shows this result?

No it is not, another reviewer caught the same error, and we have corrected the text to be 8G, thank you for also identifying this mistake.

The analysis of substrate-free bacteria revealed a remarkable distribution of CipA scaffoldins. The cellulosomes appear to cover the surface of the bacterial cell. Using our clustering algorithm, we can identify 14 clusters for the CipA-CBM3a containing at least 200 fluorescent molecules per Epsilon (Figure 7B). Is this the figure that shows this result?

This description was meant for figure 8 and we have corrected the issue in the text, thank you for pointing it out

September 25, 2025

RE: Life Science Alliance Manuscript #LSA-2025-03239-TR

Dr. Yannick Bomble
National Renewable Energy Laboratory
15013 Denver W Pkwy
Golden, CO 80401

Dear Dr. Bomble,

Thank you for submitting your revised manuscript entitled "Understanding the Dynamics of Biomass Deconstruction by the Cellulolytic Anaerobe *C. thermocellum*". As you will see, reviewers are overall satisfied. We concur with Reviewer 1 that the title and abstract do not convey the conclusions reached in this work and we make suggestions to improve these below. In addition, the text that was added to the results (lines 224-243) in response to this reviewer's question should be removed, and moreover we were disappointed that lengthy response to the this reviewer on this particular question was unhelpful. We make suggestions to improve the text below. In light of the overall support from reviewers, would be happy to publish your paper in Life Science Alliance pending final revisions necessary to meet our formatting guidelines.

- Please change the title to state the main conclusions of this work concerning cellulosome dynamics.
- Please reformulate the latter half of the abstract. Rather than describe "a clearer picture of the dynamics of cellulosome population" please state simply what the clearer view of these dynamics is. Rather than state that the methods permit testing of prevailing theories, please simply state whether the results agree or disagree with these theories.
- Please remove the text in lines 224-243, which comes after STORM and PALM were already discussed in the preceding figures, except that which is needed to properly introduce DBSCAN.
- Please upload all figure files as individual ones, including the supplementary figure files; all figure legends should only appear in the main manuscript file.
- Please add the X and Bluesky handles of your host institute/organization, as well as your own and/or one of the authors, in our system.
- The titles in both the system and the manuscript file must be consistent with each other.
- Please be sure that the authorship listing and order are correct.
- There is a name discrepancy for one of the co-authors, please correct accordingly: Dominick G. Stich in the manuscript file vs. Dominik Stich in the system.
- Please be sure to add the authors' affiliations on the title page in the manuscript file.
- It is recommended to exclude figures from the manuscript text and upload them separately.
- Please add your main and supplementary figure legends to the main manuscript text after the references section.
- Please label panels in Figure 4 to match its legend.
- LSA does not permit citation of "data not shown" in any section of the manuscript. Please remove this in line 85 where the qualitative description of prior research efforts does not merit citation of any data. Please also remove this phrase in line 307 where the underlying imaging data are represented in the referenced figure.
- Please add a Data Availability section after the Materials & Methods section. Please consult our guidelines at <https://www.life-science-alliance.org/manuscript-prep#format>.
- Please add an Author Contributions section to your main manuscript text.
- Please add the Acknowledgment section to the manuscript text.
- We encourage you to revise the figure legends for Figure S1 such that the figure panels are introduced in alphabetical order.
- Please incorporate the supplementary references into the main references list.
- Please add callouts for Figures 2A-B; 3A-D; 4A-B; 6C-G; 7A-B, F, K, M; 8A; S1A-C; S2A-C; S3A-B; S4A-B and S5A-B to your main manuscript text.
- Images appear to be duplicated between Figures 4 and 7; S3 and 3 and 6; S4 and 3. While this reuse is acceptable, you must note in the figure legends when an image is shown in another figure.
- Please remove the section title "II. Quantitative analysis of STORM and PALM optical imaging".
- Please include details on the microscopes, objectives and their NA, and other pertinent details on image acquisition and processing. For an example see: <https://doi.org/10.26508/lsa.201800028>. Please note that LSA requires all custom scripts used to generate or analyze data must made available (for example via Github).

LSA now encourages authors to provide a 30-60 second video where the study is briefly explained. We will use these videos on social media to promote the published paper and the presenting author (for examples, see

<https://docs.google.com/document/d/1-UWCfbE4pGcDdcgzcmiuJl2XMBJnxKYeqRvLLrLSo8s/edit?usp=sharing>). Corresponding or first-authors are welcome to submit the video. Please submit only one video per manuscript. The video can be emailed to contact@life-science-alliance.org

A. FINAL FILES:

B. MANUSCRIPT ORGANIZATION AND FORMATTING:

Thank you for your attention to these final processing requirements. Please revise and format the manuscript and upload materials as soon as you are able.

Sincerely,

Reviewer #1 (Comments to the Authors (Required)):

This manuscript is improved. However, in stead of doing experiments properly (automated image analyses, recapturing of images including bright field pictures), the authors do a lot of arguing. I live it at the discretion of the editor to decide. I do not see

anything wrong in the study, but it could have been done much better.

Reviewer #2 (Comments to the Authors (Required)):

The have made the corrections I requested

Reviewer #3 (Comments to the Authors (Required)):

The authors answered all the questions previously raised and the manuscript is suitable for publication in the current version.

December 15, 2025

RE: Life Science Alliance Manuscript #LSA-2025-03239-TRR

Dr. Yannick Bomble
National Renewable Energy Laboratory
15013 Denver W Pkwy
Golden, CO 80401

Dear Dr. Bomble,

Thank you for submitting your Research Article entitled "Deconstruction by *C. thermocellum* - From Microbe Mediated to Dynamic Redistribution of Cellulosomes". It is a pleasure to let you know that your manuscript is now accepted for publication in Life Science Alliance. We appreciate your attention to the several changes needed to align with our journal policies. Congratulations on this interesting work.

Your manuscript will now progress through copyediting and proofing. During proofing please make the following final changes:

- Include callouts in the main text for Figure panels S4A-B and S5A-B
- Remove the colon in the section title in line 197.

DISTRIBUTION OF MATERIALS:

Again, congratulations on a very nice paper. I hope you found the review process to be constructive and are pleased with how the manuscript was handled editorially. We look forward to future exciting submissions from your lab.

Sincerely,
